# Peeling Context from Cause for Multimodal Molecular Property Prediction

## Abstract

Deep models are used for molecular property prediction, yet they are often hard to interpret and may rely on spurious context rather than causal structure, degrading reliability under distribution shift and harming predictive performance. We introduce **CLaP**, *Causal Layerwise Peeling*, a framework that separates causal signal from context in a layerwise manner and integrates diverse graph representations of molecules. At each layer, a *causal block* performs a soft split into causal and context branches, fuses causal evidence across modalities, and *peels* batch-coupled context to concentrate on label-relevant structure, limiting shortcut signals and stabilizing layerwise refinement. Across nine molecular benchmarks, including OOD benchmarks, CLaP reliably reduces MAE and MSE relative to competitive baselines. We also obtain atom-level causal saliency maps that highlight substructures for a prediction, providing actionable guidance for targeted molecular edits. Case studies confirm the accuracy of these maps and their alignment with chemical intuition. By peeling context from cause at every layer, the model delivers predictors that are accurate and interpretable for molecular design.

## 1 Introduction

Designing molecules with desired properties is a central goal in drug discovery and materials design (Sanchez-Lengeling & Aspuru-Guzik, 2018). Graph-based deep learning is effective for property prediction (Wu et al., 2018; Hinton et al., 2006; Bengio & LeCun, 2007; Goodfellow et al., 2016). However, models often exploit spurious correlations tied to datasets or batches (Geirhos et al., 2020), which hurts reliability under distribution shift. They also provide limited substructure-level guidance (Jiménez-Luna et al., 2020), reducing their value for design. These gaps motivate predictors that separate causal signal from contextual shortcuts and yield chemically meaningful attributions.

Prior work on invariant rationales promotes causal subgraphs by enforcing invariance across *constructed* environments (Wu et al., 2022; Sui et al., 2022; Chen et al., 2022). While influential, these approaches are largely designed for classification, rely on synthetic environments whose fidelity to real data is uncertain, and often treat causality as a one-shot selection problem. We take a different route and introduce **CLaP**, *Causal Layerwise Peeling*, a layerwise framework that peels context from cause. At each layer, a learnable splitter routes features into a *causal* branch and a *context* branch, progressively *peeling* batch-coupled context and concentrating on label-relevant structure. Rather than engineering synthetic environments such as varying graph perturbations, or selecting a sparse subgraph with a predictor trained to be stable across them, we exploit the natural fluctuations of ordinary mini-batches during training as contextual variation that the model learns to ignore.

Our design is grounded in a *batch-wise invariance* principle. As batch composition changes during training, only sample intrinsic signal should keep a stable alignment with the label. We implement this with a depth-dependent correlation target and a monotonicity regularizer. The causal readout increases its within-batch Pearson correlation with the label from shallow to deep layers. A stop gradient residual objective trains the context branch on the remaining error. This enforces a clean division of labor and prevents shortcut leakage into the causal path. Building on this premise, we formulate a *regression-ready* objective that aligns the scalar causal readout with continuous labels and enforces non-decreasing correlation across depth. These design choices enable CLaP to *peel* context from cause without synthetic environments and to provide robust, interpretable predictions.

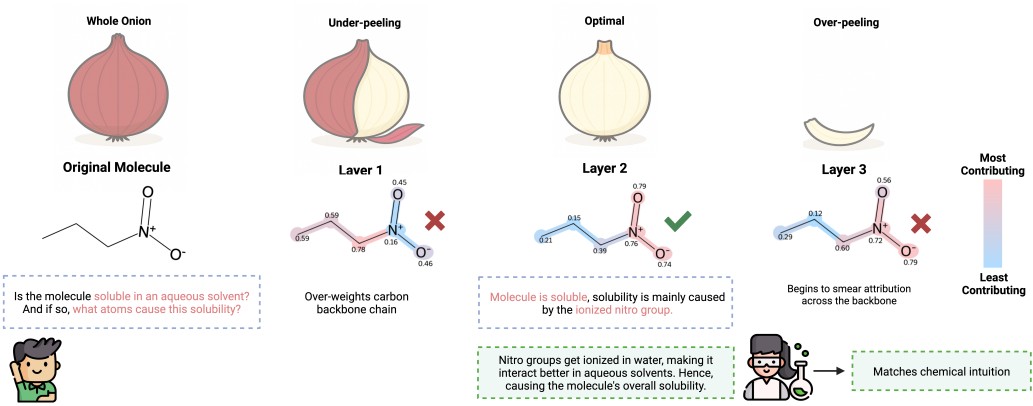

Figure 1: The original molecule in this illustrative example is iteratively "peeled" across layers. At shallow layers (layer 1), attribution underpeels and overweighs the carbon backbone. At the optimal layer, causal weights concentrate on the nitro group, yielding chemically plausible solubility drivers. Deeper peeling begins to smear attribution across the backbone, reflecting over-peeling in practice.

From a molecular-property standpoint, different representations usually capture complementary facets of the same molecule. We therefore fuse same-depth causal evidence from three views—2D topology (SMILES graphs), peptide notation (HELM (Zhang et al., 2012)), and 3D geometry. A learned gating module assigns sample-specific weights, up-weighting the modality that carries the batch-invariant signal at each layer. Gating weights yield atom-level causal saliency maps, and layerwise fusion peels away context to reveal the label-driving substructures (see Fig. 1).

Experiments on small-molecule benchmarks (ESOL, FreeSolv, Lipo) and cyclic peptides (Cy-cPeptMPDB), as well as OOD evaluations on Half_Life_Obach, Solubility, LD50_Zhu, Hepatocyte, and Microsome, show that CLaP achieves the most consistent improvements in MAE and MSE over both architecture-centric baselines (MAT, FP-GNN, MolFCL, GSL) and causality-oriented baselines (DIR, CAL, CGR) (Maziarka et al., 2019; Cai et al., 2022; Wu et al., 2022; Sui et al., 2022; Tang et al., 2025; Yin et al., 2025; Zhao et al., 2024). The atom-level causal saliency maps align with chemical intuition and, in case studies, pinpoint causal substructures.

In conclusion, our contributions are as follows:

- We propose CLaP, a multimodal, layerwise framework that splits features into *causal* and *context* branches at every layer and fuses same layer causal evidence across molecular views, reducing reliance on datasets or batches specific shortcuts, enhancing causality.

- CLaP produces causal saliency maps that highlight label-critical substructures. In case studies, these maps match chemical intuition and guide molecular design.

- Across diverse molecular property datasets, CLaP surpasses strong baselines. Its causal–context split reduces reliance on spurious context, improving robustness and accuracy.

## 2 METHODOLOGY

We first formalize a *batch-wise invariance* principle that provides the theoretical basis for our supervised causal objective. We then instantiate it with a layerwise causal–context architecture that fuses modalities and splits features per layer to enable both causal alignment and interpretability.

### 2.1 BATCH-WISE INVARIANCE

During training, each molecule is repeatedly sampled into different mini-batches across epochs, and thus encounters varying *batch contexts* defined by the co-occurring samples and their label statistics. A reliable predictor should maintain consistent alignment with the ground-truth label regardless of such batch fluctuations. This observation motivates a *batch-wise invariance* principle: only a sample-intrinsic signal can sustain stable correlation with the label under changing batch contexts.

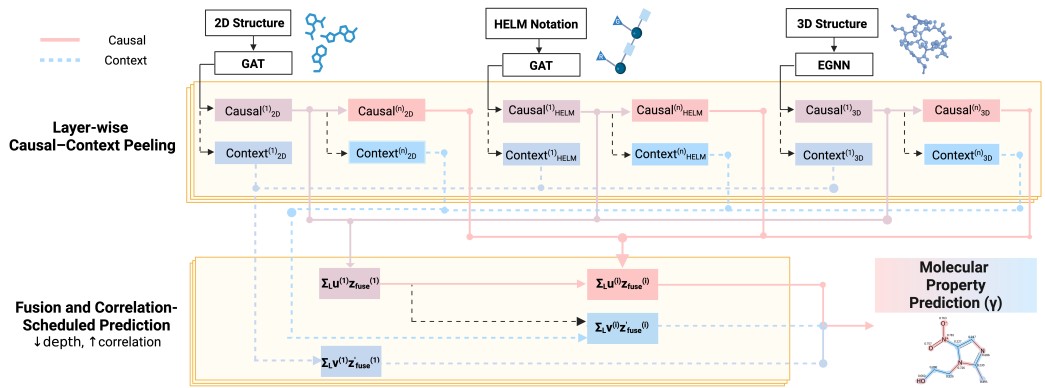

Figure 2: Overview of the causal layerwise peeling architecture for $L = 2$ layers.

**Setup.** Index mini-batches by $e \in \mathcal{E}$ over random shuffles. Assume an additive view of the label

$$y = \theta\, c(x) + \eta, \qquad \mathbb{E}[\eta \mid x] = 0, \tag{1}$$

where $c(x)$ is a sample-intrinsic, label-relevant component. Let $q(x, e)$ denote *context* features whose association with $y$ may *depend on* the batch $e$. Consider a local linearization of a layer's scalar predictor with scalars $a, b$.

$$s(x, e) = a\, c(x) + b\, q(x, e), \tag{2}$$

**Batchwise correlation.** For any batch $e$ of size $B_e$, define batch-centered variables

$$\widetilde{u}_e = u - \tfrac{1}{B_e}(\mathbf{1}^\top u)\, \mathbf{1},$$

and the within-batch Pearson correlation

$$\mathrm{Corr}_e(u, v) = \frac{\langle \widetilde{u}_e,\, \widetilde{v}_e \rangle}{\|\widetilde{u}_e\|_2\, \|\widetilde{v}_e\|_2 + \varepsilon}, \qquad \varepsilon > 0 \text{ small}. \tag{3}$$

**Invariance objective.** Batch-wise invariance requires

$$\mathrm{Corr}_e\big(s(x, e),\, y\big) \approx \rho \quad \text{for all } e \in \mathcal{E}, \text{ fixed } \rho \in [-1, 1]\,. \tag{4}$$

i.e., small dispersion of $\mathrm{Corr}_e(s, y)$ across changing batches. A convenient surrogate is

$$\min_{a, b} \sum_{e \in \mathcal{E}} \Big( \mathrm{Corr}_e\big(s(x, e), y\big) - \rho \Big)^2. \tag{5}$$

**Assumption.** We make the following assumptions:

A1. $\mathrm{Corr}_e\big(c(x), y\big) = \kappa$ is constant in $e$ (sample-intrinsic stability).

A2. $\mathrm{Corr}_e\big(q(x, e), y\big)$ varies with $e$ (batch-sensitive context).

A3. $\mathrm{Var}_e\big(c(x)\big)$ and $\mathrm{Var}_e\big(q(x, e)\big)$ are bounded away from 0.

Then any predictor of the form equation 2 that satisfies the batch-wise invariance requirement equation 4 for all $e \in \mathcal{E}$ must have $b = 0$. Equivalently, the batch-invariant solution discards $q$ and retains $c$, and $\mathrm{Corr}_e(s, y) = \mathrm{Corr}_e(c, y) = \kappa$ for all $e$. A proof for invariance is provided in Appendix A.1.

## 2.2 Layer-wise Causal–context Peeling

Our framework integrates three molecular views—2D graphs (SM), HELM notations (PE), and 3D geometries (GE). At each layer, causal blocks softly split features into *causal* and *context* paths. Causal signals are fused across modalities and aligned with the label via a depth-wise correlation schedule, while residual context is peeled and accumulated with stop-gradient. The final prediction combines the last causal readout and the accumulated residual (see Fig. 2).

**Causal block.** From the modality-specific node embeddings produced by the backbone encoders, each block at depth $\ell \in \{1, \ldots, L\}$ applies a learnable node-wise splitter that assigns every atom/group a causal gate $\alpha_c^{(\ell)} \in [0,1]$. Its complement $\alpha_t^{(\ell)} = 1 - \alpha_c^{(\ell)}$ acts as the context gate. These gates induce two branches per modality: a causal representation $z_c^{(\ell)}$ and a context representation $z_t^{(\ell)}$. For each modality at depth $\ell$, we obtain causal vectors $z_{SM}^{(\ell)}, z_{PE}^{(\ell)}, z_{GE}^{(\ell)} \in \mathbb{R}^D$ from the current block, along with the context vectors $z_{SM}^{\prime(\ell)}, z_{PE}^{\prime(\ell)}, z_{GE}^{\prime(\ell)} \in \mathbb{R}^D$. A gating network $g_F(\cdot)$ produces sample-specific soft weights $w^{(\ell)}$ over modalities via a softmax, and these weights are used to form the fused causal representation $z_{fuse}^{(\ell)}$ (and the fused context representation $z_{fuse}^{\prime(\ell)}$) at depth $\ell$.

$$w^{(\ell)} = \mathrm{softmax}\Big( g_F\big([z_{SM}^{(\ell)}; z_{PE}^{(\ell)}; z_{GE}^{(\ell)}]\big)/\tau \Big) \in \mathbb{R}^3,$$

Scalar readouts use projection heads. Applying causal head $u^{(\ell)}$ to $z_{fuse}^{(\ell)}$ yields $c^{(\ell)}$, while applying context head $v^{(\ell)}$ to $z_{fuse}^{\prime(\ell)}$ yields $t^{(\ell)}$. The causal branch is forwarded to depth $\ell+1$ (yielding $c^{(\ell+1)}$) while an additional context component $t^{(\ell+1)}$ is peeled. This recursion continues until depth $L$.

After block $L$, the per-batch scalars are stacked across layers to enable supervision and analysis.

$$C = \big[c^{(1)} \cdots c^{(L)}\big], \qquad T = \big[t^{(1)} \cdots t^{(L)}\big] \ \in \mathbb{R}^{B \times L}.$$

**Causal branch: depth-guided correlation.** The causal branch is designed to focus, layerwise, on the label-predictive signal. At each depth $\ell$, we align the fused causal scalar $c^{(\ell)}$ with the batch labels $y \in \mathbb{R}^B$ by *constraining* their Pearson correlation to follow a depth-dependent target schedule.

We first center the variables and then calculate the mini-batch Pearson correlation at depth $\ell$ with a small stabilizer $\epsilon > 0$.

$$\widetilde{c}^{(\ell)} = c^{(\ell)} - \tfrac{1}{B}(\mathbf{1}^\top c^{(\ell)})\,\mathbf{1} \qquad \widetilde{y} = y - \tfrac{1}{B}(\mathbf{1}^\top y)\,\mathbf{1},$$
$$\mathrm{corr}_\ell = \frac{\langle \widetilde{c}^{(\ell)}, \widetilde{y}\rangle}{\|\widetilde{c}^{(\ell)}\|_2\,\|\widetilde{y}\|_2 + \epsilon} \in [-1,1]. \tag{6}$$

We set a linearly increasing target correlation schedule so that deeper layers are encouraged to capture progressively finer, label-critical variation:

$$\rho_\ell = \rho_{\min} + \frac{\ell-1}{L-1}\,(\rho_{\max} - \rho_{\min}), \qquad \ell = 1, \ldots, L. \tag{7}$$

Then penalize deviations between the batchwise Pearson correlation and its target,

$$\mathcal{L}_{\mathrm{corr}} = \frac{1}{L}\sum_{\ell=1}^{L}\big(\mathrm{corr}_\ell - \rho_\ell\big)^2, \tag{8}$$

The correlation loss guides each layer toward the target depth-wise trend and concentrates causal signal while filtering out irrelevant batch-specific context noise.

To stabilize the hierarchy and prevent causal regression from $\ell$ to $\ell+1$, we further apply a monotonicity regularizer, encouraging correlations to be non-decreasing from layer to layer via a hinge penalty with margin $\gamma \geq 0$:

$$\mathcal{L}_{\mathrm{mono}} = \frac{1}{L-1}\sum_{\ell=1}^{L-1}\max\Big\{0,\ \mathrm{corr}_\ell - \mathrm{corr}_{\ell+1} + \gamma\Big\}. \tag{9}$$

Together, equation 8–equation 9 encourage a depth-wise progression. Shallow layers capture coarse, high-variance patterns, and deeper layers increasingly model causal patterns.

**Context branch: residual learning.**   The context branch isolates context and irrelevant variation that are not central to prediction. By removing these factors from the causal branch, the causal signal becomes clearer and more discriminative. At depth $\ell$, each block yields a context scalar $\boldsymbol{t}^{(\ell)}$. We aggregate these across layers, while retaining only the final causal scalar as the causal output:

$$t = \sum_{\ell=1}^{L} \boldsymbol{t}^{(\ell)} \in \mathbb{R}^B, \qquad y_{\mathrm{c}}^{\star} = \boldsymbol{c}^{(L)}. \tag{10}$$

To maintain a clear division of labor, the context branch is trained on the residual between the ground-truth label $\boldsymbol{y}$ and the final causal output, with gradients stopped through $y_{\mathrm{c}}^{\star}$ to avoid contaminating the causal pathway:

$$\mathcal{L}_{\mathrm{triv}} = \mathrm{MSE}\Big(t, \ \boldsymbol{y} - \mathrm{stopgrad}(y_{\mathrm{c}}^{\star})\Big). \tag{11}$$

This design assigns the stable, generalizable component to the causal path, whereas the context path captures the remaining residual variation, calibrating $y_{\mathrm{c}}^{\star}$ by absorbing context-specific noise that should not be attributed to the causal branch.

**Prediction and total objective.**   The model predicts by additively combining the depth-$L$ causal readout with the accumulated context correction:

$$\widehat{y} = y_{\mathrm{c}}^{\star} + t, \qquad \mathcal{L}_{\mathrm{pred}} = \mathrm{MSE}(\widehat{y}, \boldsymbol{y}). \tag{12}$$

We regularize the causal branch toward *batch-invariant*, depth-increasing alignment with the label, while training the context branch on the residual (with stop–gradient on $y_{\mathrm{c}}^{\star}$). The full objective is

$$\mathcal{L}_{\mathrm{total}} = \mathcal{L}_{\mathrm{pred}} + \underbrace{\lambda_{\mathrm{caus}} \mathcal{L}_{\mathrm{corr}}}_{\text{causal alignment}} + \underbrace{\lambda_{\mathrm{mono}} \mathcal{L}_{\mathrm{mono}}}_{\text{depth monotonicity}} + \underbrace{\lambda_{\mathrm{unif}} \mathcal{L}_{\mathrm{triv}}}_{\text{residual fit}}, \quad \lambda_{\mathrm{caus}}, \lambda_{\mathrm{mono}}, \lambda_{\mathrm{unif}} \geq 0.$$

## 3 EXPERIMENTS

### 3.1 EXPERIMENTAL SETTINGS

**Datasets.**  We evaluate CLaP on a total of nine molecular property prediction datasets, organized into two evaluation settings: in-distribution (ID) and out-of-distribution (OOD).

In the in-distribution setting, we apply random splits to four representative small-molecule and peptide benchmarks: ESOL (aqueous solubility) (Delaney, 2004), FreeSolv (hydration free energy) (Mobley & Guthrie, 2014), Lipo (octanol/water partition coefficient) (Hersey, 2015), and CycPeptMPDB (passive permeability of cyclic peptides) (Li et al., 2023).

To evaluate robustness under distribution shift, we adopt preset scaffold-based OOD splits on five additional datasets from the TDC (Huang et al., 2021): Half_Life_Obach, Solubility, LD50_Zhu, Hepatocyte, and Microsome, which jointly cover absorption, excretion, and toxicity properties.

**Baselines.** We compare CLaP with two major categories of molecular property prediction methods: (i) *architecture-centric* models that emphasize network design, including MAT (Maziarka et al., 2019) (Molecule Attention Transformer), FP-GNN (Cai et al., 2022) (Fingerprint-enhanced GNN), GSL (Graph Structure Learning) (Zhao et al., 2024), and MolFCL (fragment- and functional-group-aware contrastive learning) (Tang et al., 2025); and (ii) *causality-oriented* approaches that improve prediction by modeling causal structure or mitigating confounding effects, including DIR (Wu et al., 2022) (Discovering Invariant Rationales) and CAL (Sui et al., 2022) (Causal Attention Learning). For causality-based models originally proposed for classification, we preserve their causal mechanisms and adapt only the output layer to produce scalar regression outputs. Espacially, We include CGR (Causal Graph Learning) (Yin et al., 2025), which leverages causal intervention to enchance molecular regression. See Appendix D.2 for further implementation details.

### 3.2 MAIN RESULTS

Table 1 compares CLaP against *seven* baselines on *nine* datasets spanning both ID (ESOL, FreeSolv, Lipo, CycPeptMPDB) and OOD (Half Life, Hepatocyte, Microsome, Solubility, Toxicity) settings. We report MAE and MSE (lower is better). All results are averaged over 3 runs.

Table 1: Comparison across nine benchmarks and ablation results. Best per-dataset/metric in **bold**.

| Method | ESOL | | FreeSolv | | Lipo | | CycPeptMPDB | | Half Life | | Hepatocyte | | Microsome | | Solubility | | Toxicity | |
|---|---|---|---|---|---|---|---|---|---|---|---|---|---|---|---|---|---|---|
| | MAE | MSE | MAE | MSE | MAE | MSE | MAE | MSE | MAE | MSE | MAE | MSE | MAE | MSE | MAE | MSE | MAE | MSE |
| CAL | 0.4923 | 0.4155 | 1.1273 | 2.2452 | 0.6217 | 0.6536 | 0.3682 | 0.2202 | 1.0189 | 1.6643 | 0.9839 | 1.6163 | 0.8335 | 1.4078 | 1.0572 | 1.9931 | 0.6636 | 0.7446 |
| FP-GNN | 0.5039 | 0.5150 | 1.1561 | 2.0783 | 0.4809 | 0.4890 | 0.3487 | 0.2202 | 0.8857 | 1.5464 | 1.0836 | 1.5861 | 0.8083 | 1.2308 | 0.8383 | 1.2841 | 0.5242 | 0.5100 |
| CGR | 1.5853 | 4.0291 | 2.7268 | 11.4813 | 0.9670 | 1.5269 | 0.5792 | 0.5578 | 1.0360 | 1.6673 | 1.2371 | 1.9796 | 1.2544 | 1.9550 | 1.6741 | 4.5156 | 0.7794 | 1.0112 |
| GSL | 1.0032 | 1.2566 | 2.3476 | 4.0343 | 0.9007 | 1.1478 | 0.6590 | 0.8127 | 0.9365 | 1.3304 | 1.1045 | **1.3201** | 1.0723 | 1.2428 | 1.5883 | 1.9713 | 0.7414 | 0.9073 |
| DIR | 0.6267 | 0.6794 | 2.2420 | 7.8170 | 0.6328 | 0.7313 | 0.4243 | 0.2927 | 0.9600 | 1.6880 | 1.0780 | 1.8210 | 1.1580 | 2.1250 | 1.1890 | 4.5220 | 0.6810 | 0.8050 |
| MAT | 0.5394 | 0.4803 | 0.7680 | 1.2255 | 0.5284 | 0.4745 | 0.4129 | 0.3146 | 0.9915 | 1.5690 | 1.5660 | 2.6050 | 1.1350 | 1.5820 | 0.9250 | 1.2220 | 0.8622 | 0.9300 |
| MolFCL | 0.4660 | 0.4740 | 1.0300 | 1.9890 | **0.4330** | 0.3670 | 0.3690 | 0.2440 | 0.9210 | 1.5000 | 1.1160 | 1.7660 | 0.9430 | 1.3580 | **0.6990** | **0.9060** | 0.7530 | 1.0920 |
| w/o causal split | 0.5094 | 0.4939 | 1.2858 | 2.5719 | 0.5145 | 0.4302 | 0.3120 | 0.1834 | 0.8301 | 1.2055 | 1.0571 | 1.6711 | 0.7908 | 1.1071 | 0.7850 | 1.1157 | 0.5184 | 0.5090 |
| w/o correlation | 0.5078 | 0.4767 | 0.8779 | 1.3742 | 0.5562 | 0.5122 | 0.3215 | 0.1866 | 0.8222 | **1.1382** | 1.0230 | 1.5725 | 0.8051 | 1.0930 | 0.7835 | 1.1295 | 0.5250 | 0.5087 |
| w/o context | 0.4732 | 0.4137 | 1.0753 | 1.7338 | 0.5824 | 0.5348 | 0.3214 | 0.2037 | 0.8123 | 1.1543 | 1.0330 | 1.5965 | 0.7805 | 1.0647 | 0.7801 | 1.1035 | 0.5189 | 0.4993 |
| **CLaP** | **0.4456** | **0.3583** | **0.7020** | **0.8866** | 0.4672 | **0.3645** | **0.3056** | **0.1644** | **0.7832** | 1.1537 | **0.9520** | 1.4108 | **0.7221** | **0.9682** | 0.7725 | 1.0919 | **0.5064** | **0.4858** |

CLaP ranks **first on 7/9 datasets for MAE** and **first on 7/9 for MSE**. It is competitive on the remaining tasks: on *Lipo* CLaP attains the best **MSE** (0.3645) while MolFCL has the lowest MAE (0.4330), and on *Hepatocyte* CLaP has the lowest **MAE** (0.9520) with an MSE close to the best (1.4108 vs. 1.3201 for GSL). *Solubility* is the only dataset where MolFCL leads on both metrics.

Representative gains over the best non–CLaP baseline are substantial: on *CycPeptMPDB* CLaP reduces MAE by **12.4%** (0.3056 vs. 0.3487; FP-GNN) and MSE by **25.3%** (0.1644 vs. 0.2202; CAL/FP-GNN). On *FreeSolv* the reductions are **8.6%** MAE (0.7020 vs. 0.7680; MAT) and **27.7%** MSE (0.8866 vs. 1.2255; MAT). We also observe strong OOD gains, e.g., on *Microsome* **10.7%** MAE and **21.3%** MSE (0.7221/0.9682 vs. 0.8083/1.2308; FP-GNN) and on *Toxicity* **3.4%** MAE and **4.7%** MSE (0.5064/0.4858 vs. 0.5242/0.5100; FP-GNN). On *ESOL*, CLaP improves over the best baseline by **4.4%** MAE (0.4456 vs. 0.4660; MolFCL) and **13.8%** MSE (0.3583 vs. 0.4155; CAL). For *Half Life*, CLaP lowers MAE by **11.6%** (0.7832 vs. 0.8857; FP-GNN) and MSE by **13.3%** (1.1537 vs. 1.3304; GSL). On *Lipo*, CLaP achieves the best MSE (0.3645, **0.7%** below MolFCL's 0.3670) while trailing the MAE leader (0.4672 vs. 0.4330; MolFCL). On *Solubility*, MolFCL remains strongest (0.6990/0.9060 vs. 0.7725/1.0919).

Overall, these results indicate that *causal–context peeling*—via the correlation schedule and residual context calibration—consistently improves data efficiency and robustness across both ID and OOD regimes, with especially large gains on challenging settings such as *CycPeptMPDB* and *FreeSolv*.

## 3.3 VARIANTS AND ABLATIONS

### 3.3.1 FRAMEWORK DESIGN

Table 1 also reports ablations of core framework components. The full model integrates the depth-wise correlation schedule (Eq. equation 8) and a stop–gradient context branch (Eq. equation 11).

**w/o causal–context split.** This variant removes the correlation loss $\mathcal{L}_{\text{corr}}$, the context loss $\mathcal{L}_{\text{triv}}$, and the monotonicity penalty $\mathcal{L}_{\text{mono}}$, thereby collapsing our two-branch design. Training reduces to plain regression with $\mathcal{L}_{\text{pred}}$ (Eq. equation 12). Empirically, performance drops (e.g., on the CycPeptMPDB dataset), indicating that straightforward end-to-end fitting struggles to separate label-relevant structure and is easily distracted by spurious information.

Under the batch-wise invariance view, a predictor that mixes batch-dependent context $q(x, e)$ with intrinsic signal $c(x)$ yields $\text{Corr}_e(s, y)$ that fluctuates with $e$. Removing the split collapses the losses $\mathcal{L}_{\text{corr}}, \mathcal{L}_{\text{triv}}, \mathcal{L}_{\text{mono}}$, allowing $b \neq 0$ in Eq. equation 2, contradicting the invariant solution driven by Eq. equation 5. The performance drop reflects increased reliance on batch-sensitive shortcuts.

**w/o correlation schedule.** We omit the correlation loss $\mathcal{L}_{\text{corr}}$ and (Eq. equation 8), which disables the target schedule for $\rho$. The depth schedule $\{\rho_\ell\}$ (Eq. equation 7) and $\mathcal{L}_{\text{corr}}$ (Eq. equation 8) specify a curriculum from coarse to decisive label-relevant factors.

Without this schedule, the stack lacks directional guidance and drifts toward batch-specific shortcuts. It fits residual patterns with little label signal, which hurts generalization. Keeping the monotonicity

term $\mathcal{L}_{\mathrm{mono}}$ is not enough because the non-decreasing constraint lacks direction and cannot steer which features to keep or discard. In the full model the schedule works as a curriculum that allocates capacity to increasingly predictive components and focuses deeper layers on label relevant variation.

**w/o context branch.** We ablate the context branch and predict solely from the final-layer causal output $y_{\mathrm{c}}^{\star}$. This forces residual context into the causal branch and eliminates residual calibration of environment-driven variation, causing *leakage* of non-causal factors and a performance drop. See Appendix A.2 for a formal derivation and risk decomposition.

For instance, in a testing example with ground-truth permeability $y = -0.80$, we observe $y_{\mathrm{c}}^{\star} = -0.78$ and $t = -0.01$, yielding a final prediction $\widehat{y} = y_{\mathrm{c}}^{\star} + t = -0.79$. This decomposition reflects how $y_{\mathrm{c}}^{\star}$ captures the stable, mechanism-driven component, while $t$ calibrates a small, context-dependent bias. Without the context head, however, such residual calibration becomes unavailable. The model is forced to encode both signals into $y_{\mathrm{c}}^{\star}$, entangling causal and contextual information.

### 3.3.2 CAUSAL INVARIANCE UNDER RE-BATCHING

**Setup as intervention.** To evaluate the robustness of the learned representations under distributional shifts, we treat the test-time batch setup as an intervention variable. We fix the trained model and *intervene* on the test-time environment $E$ by changing how the same test examples are partitioned into mini-batches: $E = (B, \mathrm{shuffle})$ with $B \in \{8, 16, 32\}$ and with/without shuffling. This realizes $\mathrm{do}(E = e)$ that perturbs batch boundaries and co-occurrences while holding $c(x)$ fixed. In the notation of Sec. 2.1, such interventions modify the batch-dependent context $q(x, E)$ in the local predictor $s(x, E) = ac(x) + bq(x, E)$ (Eq. equation 2) but leave the sample-intrinsic signal $c(x)$ unchanged. Our learning mechanism (depth-wise correlation targets equation 6–equation 8) is designed to drive $b \to 0$, i.e., to make the causal readout rely on $c(x)$ and be invariant to $\mathrm{do}(E = e)$.

Table 2: Test $R^2$ under $\mathrm{do}(E)$ via re-batching.

|  | B=8 | B=16 | B=32 |
|---|---|---|---|
| $R^2$ (no) | 0.7172 | 0.7282 | 0.7219 |
| $R^2$ (shuf) | 0.7120 | 0.7165 | 0.7184 |

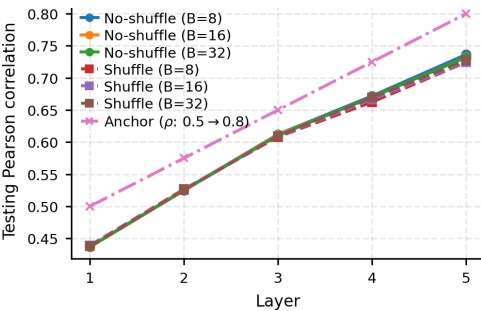

Figure 3: Environment interventions via re-batching.

For each intervention we compute *global* correlations between $c$ and $y$ over the test set. The correlation–depth curves (Fig. 3) remain monotonic and track the anchors for all $B$ and shuffling choices. (Table 2) Final-layer correlations lie in $[0.7120, 0.7282]$ with total spread $\le 0.0162$. Since interventions change $E$ while keeping the examples fixed, this stability indicates that the learned alignment is insensitive to environment construction, as predicted by the batch-wise invariance principle supporting our claim that the predictor relies primarily on the sample-intrinsic component $c(x)$.

### 3.3.3 PEELING DESIGN

Table 3: Layers and $\rho_{\mathrm{max}}$ vs. performance on CycPeptMPDB. Best indicated in **Bold**.

|  | *Peeling Depth (L)* | | *Peeling Strength ($\rho_{\mathrm{max}}$)* | | |
|---|---|---|---|---|---|
| $L$ | MAE | MSE | $\rho_{\mathrm{max}}$ | MAE | MSE |
| 3 | 0.3265 | 0.1903 | 0.60 | 0.3131 | 0.1741 |
| **5** | **0.3056** | **0.1644** | 0.70 | 0.3162 | 0.1733 |
| 7 | 0.3296 | 0.2020 | **0.80** | **0.3056** | **0.1644** |
| 9 | 0.3297 | 0.1875 | 0.90 | 0.3133 | 0.1834 |

We analyze how peeling depth $L$ (number of causal blocks) and strength $\rho_{\mathrm{max}}$ (maximum target correlation) shape performance. Pushing either to extremes yields *under-* or *over*-peeling.

**Peeling depth.** We vary the number of causal blocks $L \in \{3, 5, 7, 9\}$ and observe a U-shaped response in MAE/MSE on CycPeptMPDB in the upper block of Table 3. With too few blocks

($L=3$), the hierarchy *under-peels*. It lacks capacity to progressively strip context, leaving causal and context signals entangled and hurting accuracy. With too many blocks ($L=7, 9$), the hierarchy *over-peels*. The longer optimization path increases variance and can misroute label-relevant patterns into the context stream despite the monotonicity regularizer, degrading generalization. An interior optimum at $L=5$ aligns with a bias–variance view: depth must be large enough to separate signals but not so large that true causal evidence is shaved off together with context.

**Peeling strength.** We also examine the effect of the target-correlation parameter $\rho_{max} \in \{0.60, 0.70, 0.80, 0.90\}$ on CycPeptMPDB in the lower block of Table 3. Smaller targets (0.60–0.70) *under-peel*. The causal branch is not pushed hard enough, so contextual residue persists and errors rise. A moderate target (0.80) yields the best trade-off. Pursuing near-perfect within-batch correlation (0.90) *over-peels*. To satisfy an aggressive target, the model exploits batch-coupled features that should remain context, increasing variance and hurting generalization. Appendix A.3 provides additional evidence that an overly large target correlation causes context leakage.

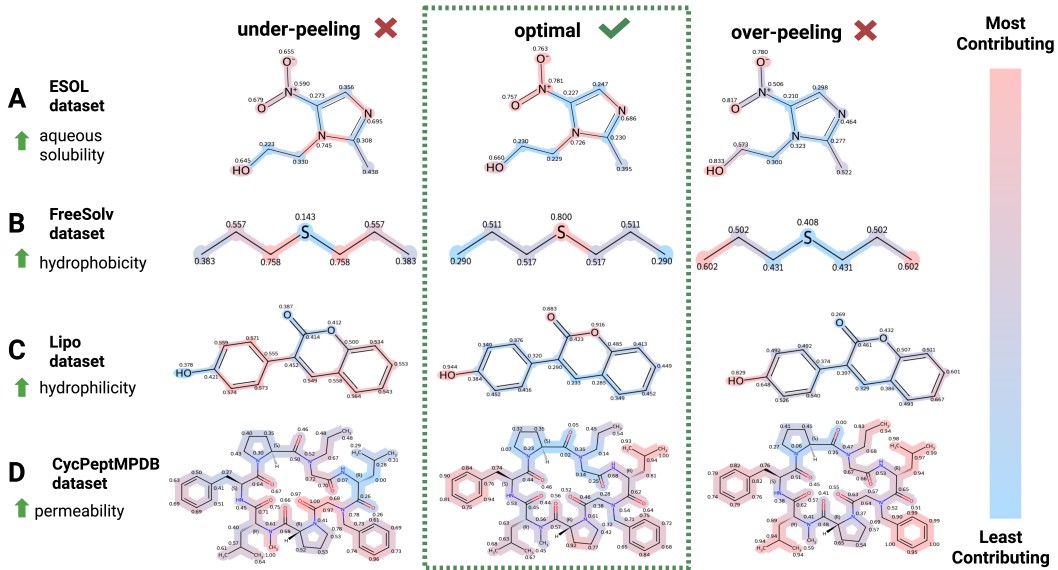

Figure 4: Layer-wise attribution maps in the multimodality setting across four benchmark datasets. Each row corresponds to a dataset and the molecular property of interest, indicated by the green arrow on the left. Three regimes are shown: under-peeling (left), optimal (center), and over-peeling (right). Atom-level contributions are highlighted with red atoms representing features most attributed to increasing the target property and blue atoms representing less contributing features. Optimal peeling yields clearer and more interpretable attribution compared to under- and over-peeling.

### 3.4 CASE STUDY

Each causal block applies node-wise gating on atoms, producing $\alpha_i^{(\ell,m)} \in [0, 1]$ for atom/group $i$ in modality $m \in \{SM, PE, GE\}$ at depth $\ell$. We take the final-layer weight as the causal score, $\pi_i^{(m)} = \alpha_i^{(L,m)} \in [0, 1]$, and render *causal saliency maps* $\{\pi_i^{(m)}\}$ to inspect learned behavior.

Higher $\pi_i^{(m)}$ indicates atoms routed through the causal branch at the final layer. Figure 4 compares the under-peeling, optimal-peeling, and over-peeling settings. In the optimal case, saliency concentrates on chemically meaningful moieties, *consistent with chemical intuition*.

At the optimal depth, CLaP concentrates attribution on chemically coherent drivers per task.

Across the three datasets, **A. ESOL** highlights that polar nitro groups and heteroatoms consistently dominate, reflecting their strong role in hydrogen bonding and aqueous solubility (especially in water); **B. FreeSolv** shows that hydrocarbon segments together with the thio-linker carry the highest weights, aligning with sulfur's tendency to raise local hydrophobicity and thus overall lipophilicity; and **C. Lipo** indicates that phenolic OH substituents and ring heteroatoms elevate hydrophilicity

while the extended aromatic surface is down-weighted, capturing the trade-off between polar handles and nonpolar bulk consistent with established lipophilicity trends. Finally, in **D. CycPeptMPDB dataset**, contiguous aromatic and aliphatic side chains form lipophilic patches that promote passive permeation, whereas backbone hydrogen-bond donors and acceptors contribute less, consistent with the desolvation cost of membrane crossing. This pattern reflects the intended peel—the shallow layers under-localize the signal, the deep layers over-peel and smear attribution, and the middle layers isolate chemically relevant substructures that drive the property.

## 3.5 COUNTERFACTUAL STUDY

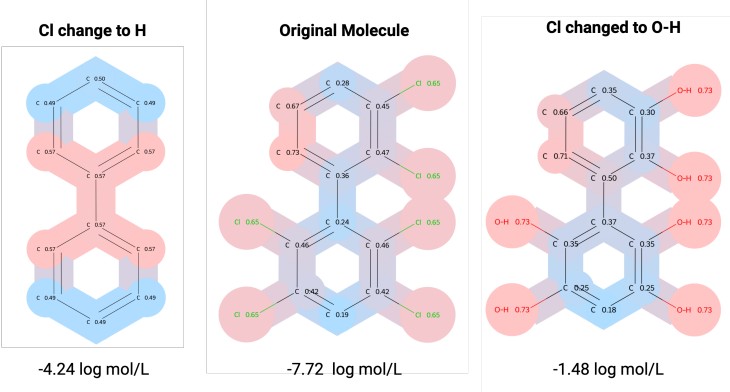

Figure 5: Counterfactual explanation for ESOL solubility prediction produced by CLaP. Left: context-level attribution highlighting the central hydrophobic scaffold as driving low solubility. Middle: atom-level attributions for the original chlorinated molecule with a predicted solubility of $-7.72$ log mol/L. Right: counterfactual edit obtained by replacing all Cl substituents with O–H groups, which receive strong positive attributions and increase the predicted solubility to $-1.48$ log mol/L, illustrating that CLaP's explanations align with chemically meaningful modifications.

To further validate the learned causal weights, we perform a counterfactual experiment by substituting the highest-weight atoms with more soluble functional groups (Figure 5), and examine both solubility shifts and causal-weight responses. In the original molecule, the `Cl` atoms receive the highest causal scores, indicating that the model identifies them as primary solubility determinants. Replacing `Cl` with `H` removes these high-impact sites and leads to a marked increase in solubility. In contrast, substituting `Cl` with an `O-H` group preserves the same causal positions but reverses their chemical effect, producing an even larger increase. These counterfactual edits demonstrate that the model not only locates scaffold-level "causal sites," but also correctly predicts both the *direction* and *magnitude* of property changes induced by specific chemical substitutions, reflecting true counterfactual reasoning.

## 4 RELATED WORK

**Graph causal inference.** Recent work seeks to improve robustness and interpretability in graph learning by separating label-relevant causal structure from spurious contextual signals. DIR selects invariant rationales (subgraphs that remain predictive across environments) and then predicts with the remainder (Wu et al., 2022). CIGA instead samples subgraphs via a learnable generator and enforces risk invariance through an information bottleneck (Chen et al., 2022).

In molecular settings, environment construction is often simulated by adding or substituting functional groups and treating these variants as separate environments (Li et al., 2025). However, such manipulations may be chemically unrealistic and can exacerbate the distribution gap between training and testing data. CAL alleviates shortcut learning and improves interpretability (Sui et al., 2022), yet most prior work focuses on *classification*. Regression is inherently more challenging: the model

must precisely localize causal substructures rather than rely on class-level artifacts or thresholds. Most recently, CGL extends causal graph learning to regression by jointly modeling causal and confounding subgraphs, with a particular focus on molecular property prediction (Yin et al., 2025).

However, instead of crafting synthetic environments, our proposed CLaP achieves batch-wise invariance via the natural mini-batch reshuffling that occurs during training, yielding stable within-batch correlations across depths and better causal capture under re-batching.

**Molecular property prediction.** Graph neural networks have become a cornerstone for molecular property prediction (Gilmer et al., 2017; Wu et al., 2018), achieving strong performance by learning expressive molecular representations. Beyond classical message passing, recent architectures incorporate more complex structural priors: transformer-based models such as MAT encode geometric biases via inter-atomic distances (Maziarka et al., 2019), while multimodal frameworks fuse chemical language with graph structure for richer cross-domain context (Rollins et al., 2024).

Despite these advances, most models behave as black boxes, capturing correlations without disentangling true causal factors—limiting interpretability and thus their usefulness in rational molecular design (Jiménez-Luna et al., 2020). To address this, several recent works aim to enhance representation quality with structural insights or chemical priors. GSL-MPP jointly optimizes molecular features by combining GNN-based intramolecular learning with graph structure learning over an inter-molecular similarity graph, improving predictive performance through relational signals (Zhao et al., 2024). MolFCL further incorporates fragment–reaction–aware contrastive learning to obtain chemically robust encodings, and introduces functional-group– and atom-level prompt learning to inject chemical knowledge and interpretability (Tang et al., 2025).

## 5 DISCUSSION

We presented CLaP, a layerwise causal–context peeling framework that realizes batch-wise invariance, improving predictive accuracy and delivering atom-level, chemically consistent explanations.

Looking ahead, this framework can expand in both scope and mechanics. Specifically, it extends beyond scalar regression to classification by swapping scalar label supervision for within-class correlation objectives, while preserving depthwise alignment. Furthermore, it is not confined to molecular graphs and can be embedded in transformer backbones to route tokens into causal and context paths, yielding token level causal maps across modalities and domains. These extensions aim to better capture underlying causal structure and enable more faithful problem analysis.

## REPRODUCIBILITY STATEMENT

Implementation details and hyperparameters are in Appendix D.1. The supplementary material includes a code archive with configs, preprocessing, data loaders, architectures, and evaluations.

## ETHICS STATEMENT

We adhere to ICLR Code of Ethics. No human subjects or personally identifiable data are involved. To reduce misuse risk, we release benchmarking code and models only for these datasets and provide no synthesis or activity oracles. No conflicts of interest or sensitive sponsorships are present.

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

APPENDIX

# A  THEORETICAL ANALYSIS

## A.1  BATCH-WISE INVARIANCE ANALYSIS

We use the batch-centered correlation in Eq. equation 3 and the local form $s(x,e) = a\,c(x) + b\,q(x,e)$ in Eq. equation 2. Define within-batch moments

$$\sigma_{c,e}^2 = \mathrm{Var}_e(c), \quad \sigma_{q,e}^2 = \mathrm{Var}_e(q), \quad \sigma_{y,e}^2 = \mathrm{Var}_e(y),$$
$$\kappa = \mathrm{Corr}_e(c,y), \quad \alpha_e = \mathrm{Corr}_e(q,y), \quad \rho_e = \mathrm{Corr}_e(c,q)$$

with $\kappa$ constant in $e$ by (A1). Set

$$A_e = a\,\sigma_{c,e}, \qquad B_e = b\,\sigma_{q,e}.$$

**Closed form for** $\mathrm{Corr}_e(s,y)$**.** Starting from the definition,

$$\mathrm{Corr}_e(s,y) = \frac{\mathrm{Cov}_e(s,y)}{\sqrt{\mathrm{Var}_e(s)\,\mathrm{Var}_e(y)}} \tag{13}$$

$$= \frac{\mathrm{Cov}_e(ac+bq,\,y)}{\sqrt{\mathrm{Var}_e(ac+bq)\,\mathrm{Var}_e(y)}}$$

$$= \frac{a\,\mathrm{Cov}_e(c,y) + b\,\mathrm{Cov}_e(q,y)}{\sqrt{\mathrm{Var}_e(ac+bq)}\,\sqrt{\mathrm{Var}_e(y)}}$$

$$= \frac{a\,\sigma_{c,e}\,\sigma_{y,e}\,\kappa \;+\; b\,\sigma_{q,e}\,\sigma_{y,e}\,\alpha_e}{\sqrt{a^2\sigma_{c,e}^2 + b^2\sigma_{q,e}^2 + 2ab\,\sigma_{c,e}\sigma_{q,e}\,\rho_e}\;\sigma_{y,e}} \quad \Big(\mathrm{Cov}_e(u,v) = \sigma_{u,e}\sigma_{v,e}\mathrm{Corr}_e(u,v)\Big)$$

$$= \frac{A_e\,\kappa + B_e\,\alpha_e}{\sqrt{A_e^2 + B_e^2 + 2A_eB_e\rho_e}} =: \frac{A_e\kappa + B_e\alpha_e}{D_e}. \tag{14}$$

Here $D_e = \sqrt{A_e^2 + B_e^2 + 2A_eB_e\rho_e} = \sigma_{s,e} > 0$ is the within-batch standard deviation of $s$.

**A monotonicity observation** $(\partial_{\alpha_e}\mathrm{Corr}_e)$**.** For fixed $e$, $A_e, B_e, \rho_e$ do not depend on $\alpha_e$, hence from equation 14

$$\frac{\partial}{\partial \alpha_e}\,\mathrm{Corr}_e(s,y) = \frac{\partial}{\partial \alpha_e}\,\frac{A_e\kappa + B_e\alpha_e}{D_e} = \frac{B_e}{D_e}. \tag{15}$$

Therefore if $B_e \neq 0$ (equivalently $b \neq 0$), $\mathrm{Corr}_e(s,y)$ is strictly monotone in $\alpha_e$.

**Consequence under exact invariance** $(\mathrm{Corr}_{e_1} = \mathrm{Corr}_{e_2})$**.** Assume there exist batches $e_1, e_2$ with $\sigma_{c,e_1} = \sigma_{c,e_2}$, $\sigma_{q,e_1} = \sigma_{q,e_2}$, $\rho_{e_1} = \rho_{e_2}$, but $\alpha_{e_1} \neq \alpha_{e_2}$. Then $A_{e_1} = A_{e_2}$, $B_{e_1} = B_{e_2}$, $D_{e_1} = D_{e_2}$, and

$$\mathrm{Corr}_{e_1}(s,y) - \mathrm{Corr}_{e_2}(s,y) = \frac{B_{e_1}}{D_{e_1}}\big(\alpha_{e_1} - \alpha_{e_2}\big).$$

If $\mathrm{Corr}_e(s,y)$ is identical across batches (zero dispersion in Eq. equation 4), the right-hand side must be 0. Since $\alpha_{e_1} \neq \alpha_{e_2}$, this forces $B_{e_1} = 0$, hence $b = 0$ because $\sigma_{q,e_1} > 0$ by (A3).

**Approximate invariance via a first-order expansion.** Let $\delta_e(b) := \mathrm{Corr}_e(s,y) - \rho$, where $\rho$ is the target in Eq. equation 4. Write $\mathrm{Corr}_e(s,y) = \Phi_e(B_e)$ with $\Phi_e(B) = \dfrac{A_e\kappa + B\alpha_e}{\sqrt{A_e^2 + B^2 + 2A_eB\rho_e}}$. By the quotient rule,

$$\Phi_e'(B) = \frac{\alpha_e D_e - (A_e\kappa + B\alpha_e)D_e'}{D_e^2}, \qquad D_e' = \frac{B + A_e\rho_e}{D_e}.$$

Evaluating at $B = 0$ (so $D_e = |A_e|$) gives

$$\left.\frac{\partial}{\partial B}\,\mathrm{Corr}_e(s,y)\right|_{B=0} = \Phi_e'(0) = \frac{\alpha_e A_e^2 - A_e\kappa(A_e\rho_e)}{|A_e|^3} = \frac{\alpha_e - \kappa\rho_e}{|A_e|}. \tag{16}$$

Since $B_e = b\,\sigma_{q,e}$, the chain rule yields

$$\frac{\partial}{\partial b}\,\mathrm{Corr}_e(s,y)\bigg|_{b=0} = \sigma_{q,e}\,\frac{\partial}{\partial B}\,\mathrm{Corr}_e(s,y)\bigg|_{B=0} = \frac{\sigma_{q,e}}{|A_e|}\,(\alpha_e - \kappa\rho_e). \tag{17}$$

Therefore, for small $b$,

$$\delta_e(b) = \delta_e(0) + b\,\Gamma_e + o(b), \qquad \Gamma_e = \frac{\sigma_{q,e}}{|A_e|}\,(\alpha_e - \kappa\rho_e). \tag{18}$$

If $\sum_e \Gamma_e^2 > 0$ (which holds whenever $\alpha_e$ varies across batches by (A2) and $q$ is not batchwise collinear with $c$), minimizing the dispersion surrogate $\sum_e \delta_e(b)^2$ in Eq. equation 5 yields the closed-form

$$b^\star = -\,\frac{\sum_e \delta_e(0)\,\Gamma_e}{\sum_e \Gamma_e^2} \;+\; o(1). \tag{19}$$

**Invariance conclusion.** At $b = 0$, from equation 14 we have

$$\mathrm{Corr}_e(s,y)\big|_{b=0} = \frac{A_e\kappa}{|A_e|}, \quad \Rightarrow \quad \delta_e(0) = \frac{A_e\kappa}{|A_e|} - \rho.$$

With the natural choice $\rho = \kappa$ and $a \geq 0$ (so $A_e \geq 0$), we get $\delta_e(0) = 0$ for all $e$. Moreover, from equation 18 at $b^\star$ we have $\delta_e(b^\star) = b^\star\Gamma_e + r_e(b^\star)$ with $\sum_e r_e(b^\star)^2 = o((b^\star)^2)$. Taking the $\ell_2$-norm and using Cauchy–Schwarz,

$$|b^\star|\,\|\Gamma\|_2 \;\leq\; \|\delta(b^\star)\|_2 + \|r(b^\star)\|_2 \;=\; o(|b^\star|)\,,$$

and since $\|\Gamma\|_2 > 0$ by (A2)–(A3), it follows that $b^\star \Rightarrow 0$ as the model gradually reduces the contribution of the context component. Hence in the limit the predictor discards the batch-dependent $q$ (i.e., $b^\star \Rightarrow 0$ so $s \approx a\,c$), so

$$\mathrm{Corr}_e(s,y) \;\approx\; \mathrm{Corr}_e(c,y) = \kappa \qquad \text{for all } e.$$

## A.2 Context Branch Analysis

With squared loss and the stop–gradient in Eq. equation 11, training $t$ treats $y_{\mathrm{c}}^\star$ as a constant with respect to its parameters, so the context head solves the conditional least-squares problem

$$t^\star(x) = \arg\min_t \; \mathbb{E}\big[(y - y_{\mathrm{c}}^\star - t(x))^2 \mid x\big] = \mathbb{E}\big[y - y_{\mathrm{c}}^\star \mid x\big].$$

That is, the *best* residual correction at input $x$ is the *conditional mean* of the residual.

Let $r := y - y_{\mathrm{c}}^\star$. By the definition of MSE risk $\mathcal{R}(f) = \mathbb{E}[(y - f(x))^2]$,

$$\mathcal{R}(y_{\mathrm{c}}^\star) = \mathbb{E}[r^2].$$

Applying the Law of total expectation and conditional variance decomposition with $t \equiv 0$ gives the decomposition of the risk of $y_{\mathrm{c}}^\star$:

$$\mathcal{R}(y_{\mathrm{c}}^\star) = \mathbb{E}[r^2] = \mathbb{E}\big[\mathrm{Var}(r \mid x)\big] + \mathbb{E}\big[(\mathbb{E}[r \mid x])^2\big],$$

which splits the error into (i) irreducible noise $\mathbb{E}[\mathrm{Var}(r \mid x)]$ and (ii) predictable bias $\mathbb{E}[(\mathbb{E}[r \mid x])^2]$.

Plugging $t^\star(x) = \mathbb{E}[r \mid x]$ into the predictor $\widehat{y} = y_{\mathrm{c}}^\star + t$ removes the predictable bias:

$$\mathcal{R}(y_{\mathrm{c}}^\star + t^\star) = \mathbb{E}\Big[\mathrm{Var}(r \mid x) + (\mathbb{E}[r \mid x] - t^\star(x))^2\Big] \overset{t^\star=\mathbb{E}[r|x]}{=\!=} \mathbb{E}\big[\mathrm{Var}(r \mid x)\big].$$

So adding $t^\star$ strictly reduces risk by $\mathbb{E}[(\mathbb{E}[r \mid x])^2] \geq 0$ whenever the residual is partially predictable from $x$.

## A.3 TARGET-CORRELATION ANALYSIS

Assume the additive label model

$$y = \theta\, c(x) + \eta, \qquad \mathbb{E}[\eta \mid x] = 0, \quad \mathrm{Var}(\eta) > 0, \quad \theta \geq 0 \tag{20}$$

Then

$$\mathrm{Corr}(c, y) = \frac{\mathrm{Cov}(c, y)}{\sqrt{\mathrm{Var}(c)\mathrm{Var}(y)}} = \frac{\theta\, \mathrm{Var}(c)}{\sqrt{\mathrm{Var}(c)\, (\theta^2 \mathrm{Var}(c) + \mathrm{Var}(\eta))}} =: \kappa \ < \ 1. \tag{21}$$

Any batch-invariant predictor using only $c$ has $s(x, e) = a\, c(x)$ and thus $\mathrm{Corr}_e(s, y) = \kappa$ for all $e$.

Let $s(x, e) = a\, c(x) + b\, q(x, e)$ as in Eq. equation 2, and define

$$\sigma_{c,e}^2 = \mathrm{Var}_e(c), \quad \sigma_{q,e}^2 = \mathrm{Var}_e(q), \quad \kappa = \mathrm{Corr}_e(c, y)\,, \quad \alpha_e = \mathrm{Corr}_e(q, y), \ \rho_e = \mathrm{Corr}_e(c, q), \ A_e = a\, \sigma_{c,e}$$

From Eq. equation 14,

$$\mathrm{Corr}_e(s, y) = \frac{A_e\, \kappa + b\, \sigma_{q,e}\, \alpha_e}{\sqrt{A_e^2 + b^2 \sigma_{q,e}^2 + 2 A_e b \sigma_{q,e} \rho_e}}.$$

For the dispersion surrogate $\mathcal{L}_\rho(a, b) = \sum_e \left(\mathrm{Corr}_e(s, y) - \rho_\star\right)^2$ with target $\rho_\star \in (0, 1)$, the first-order sensitivity at the invariant point $b = 0$ is (Appendix A.1)

$$\frac{\partial}{\partial b}\, \mathrm{Corr}_e(s, y)\bigg|_{b=0} = \frac{\sigma_{q,e}}{|A_e|}\, (\alpha_e - \kappa\, \rho_e) \ =: \Gamma_e, \qquad \frac{\mathrm{d}}{\mathrm{d}b}\mathcal{L}_\rho\bigg|_{b=0} = 2(\kappa - \rho_\star) \sum_e \Gamma_e. \tag{22}$$

Under (A1)–(A3) and the mild non-collinearity $\exists e : \alpha_e \neq \kappa\rho_e$, the sum $\sum_e \Gamma_e$ is generically nonzero. If $\rho_\star > \kappa$ then $\kappa - \rho_\star < 0$, so $\mathrm{d}\mathcal{L}_\rho/\mathrm{d}b|_{b=0} \neq 0$ and $b = 0$ is not a local minimizer. Hence any optimizer of $\mathcal{L}_\rho$ satisfies $|b| > 0$: the predictor must recruit the batch-dependent $q(x, e)$ to reduce the surrogate when the target exceeds the invariant limit $\kappa$.

Moreover, writing $\delta_e(b) = \mathrm{Corr}_e(s, y) - \rho_\star$ and expanding at $b = 0$ gives

$$\delta_e(b) = (\kappa - \rho_\star) + b\, \Gamma_e + o(b), \quad \Rightarrow \quad b^\star = \frac{(\rho_\star - \kappa) \sum_e \Gamma_e}{\sum_e \Gamma_e^2} \ + \ o(1).$$

Thus the leakage magnitude grows linearly with the target gap $(\rho_\star - \kappa)$ and is controlled by the batch statistics through $\{\Gamma_e\}$.

# B    ADDITIONAL EXPERIMENTS

## B.1    ADDITIONAL ABLATION STUDY ON FRAMEWORK DESIGN

Table 4: Additional ablation study on framework design across four datasets. Best indicated in **Bold**.

| Setting | ESOL | | FreeSolv | | Lipo | | CycPeptMPDB | |
|---|---|---|---|---|---|---|---|---|
| | MAE | MSE | MAE | MSE | MAE | MSE | MAE | MSE |
| Average causal layer | 0.4798 | 0.4377 | 0.9612 | 1.5843 | 0.5621 | 0.5041 | 0.3227 | 0.1859 |
| No mono penalty | 0.5692 | 0.5740 | 1.0241 | 1.7210 | 0.5288 | 0.4590 | 0.3137 | 0.1803 |
| **CLaP** | **0.4456** | **0.3583** | **0.7020** | **0.8866** | **0.4672** | **0.3645** | **0.3056** | **0.1644** |

**w/o monotonicity constraint.** The monotonicity penalty $\mathcal{L}_{\mathrm{mono}}$ (Eq. equation 9) complements the correlation loss by coupling adjacent depths, encouraging non-decreasing label–correlation as depth increases. This depthwise ordering regularizes training, especially for deeper causal blocks, by discouraging oscillations and preventing later layers from overwriting gains established earlier.

Ablating $\mathcal{L}_{\mathrm{mono}}$ weakens layerwise refinement of causal features, leading to a modest yet consistent performance drop. Monotonicity regularizer improves hierarchical stability and makes the layerwise specialization more reliable.

**Average causal across layers.** This ablation probes the depthwise correlation design. The targets $\{\rho_\ell\}$ increase with depth (Eq. equation 7) and, together with $\mathcal{L}_{\mathrm{corr}}$ and $\mathcal{L}_{\mathrm{mono}}$ (Eqs. equation 8, equation 9), train the deepest causal block to become progressively more label–relevant. Here we replace the final-layer causal output with a uniform average of $\{c^{(\ell)}\}_{\ell=1}^{L}$. Given $\mathrm{corr}_1 \le \cdots \le \mathrm{corr}_L$ under $\mathcal{L}_{\mathrm{corr}} + \mathcal{L}_{\mathrm{mono}}$, averaging $\bar{c} = \frac{1}{L}\sum_\ell c^{(\ell)}$ mixes in low-correlation early representations. For centered variables, $\mathrm{Corr}(\bar{c}, y) \le \mathrm{Corr}(c^{(L)}, y)$, with strict inequality unless all $c^{(\ell)}$ are colinear with $c^{(L)}$. Thus using $c^{(L)}$ is theoretically favored.

In our experiments, averaging across layers undermines depthwise specialization and mixes early-layer representations that retain substantial context information into the final predictor, contaminating the causal signal and degrading performance (e.g. MSE: $0.164 \to 0.185$). These results validate the design: the last causal layer concentrates the causal signal, while earlier layers serve as progressively coarser precursors.

Together, the correlation schedule shapes *what* each depth should capture; the monotonicity term enforces *how* the hierarchy evolves; the context branch determines *where* non-causal residuals go.

## B.2    MULTI-MODALITY DESIGN

Table 5: Effect of causal peeling under *unimodal* vs. *multimodal* settings across four datasets. Best per-dataset/metric in **bold**.

| Setting | ESOL | | FreeSolv | | Lipo | | CycPeptMPDB | |
|---|---|---|---|---|---|---|---|---|
| | MAE | MSE | MAE | MSE | MAE | MSE | MAE | MSE |
| **Multimodal (w split)** | **0.4456** | **0.3583** | **0.7020** | **0.8866** | **0.4672** | **0.3645** | **0.3056** | **0.1644** |
| Multimodal (w/o split) | 0.5094 | 0.4939 | 1.2858 | 2.5719 | 0.5145 | 0.4302 | 0.3120 | 0.1834 |
| 2D Graph (w split) | 0.9453 | 1.5993 | 1.1267 | 1.9876 | 0.5171 | 0.4506 | 0.4068 | 0.2889 |
| 2D Graph (w/o split) | 0.9631 | 1.6931 | 1.2003 | 2.4157 | 0.5230 | 0.4504 | 0.4077 | 0.2884 |
| 3D Geometry (w split) | 0.5322 | 0.5235 | 1.0640 | 1.8055 | 0.6412 | 0.6519 | 0.4221 | 0.3102 |
| 3D Geometry (w/o split) | 0.5219 | 0.5036 | 1.0721 | 1.9262 | 0.6260 | 0.6449 | 0.4244 | 0.3071 |
| HELM (w split) | – | – | – | – | – | – | 0.3310 | 0.2023 |
| HELM (w/o split) | – | – | – | – | – | – | 0.3194 | 0.1910 |

In this section, we assess how the causal–context split behaves under different modality configurations. CycPeptMPDB provides three complementary views per sample (SM, PE, GE), whereas ESOL, FreeSolv, and Lipo provide two views (SM and GE). Table 5 compares unimodal variants with multimodal fusion, each trained with and without the split.

With two or more modalities, enabling the split generally yields better fit and stability across datasets. For instance, MSE decreases on CycPeptMPDB from 0.183 to 0.164 and on ESOL from

---

**Algorithm 1:** Layer-wise causal–context peeling

---

**Input** : multi-view data $(x_{\mathrm{SM}}, x_{\mathrm{PE}}, x_{\mathrm{GE}})$ with label $y$; depth $L$; epochs $E$; batch size $B$; correlation targets $\{\rho_\ell\}$; weights $\lambda_{\mathrm{caus}}, \lambda_{\mathrm{mono}}, \lambda_{\mathrm{unif}}$.
**Output:** trained predictor $\widehat{y} = c^\star + t$.
**for** $e = 1$ **to** $E$ **do**
    **for** *each mini-batch* **do**
        Encode inputs into embeddings.
        **for** $\ell = 1$ **to** $L$ **do**
            Split into causal/context parts, fuse across modalities, get scalars $c^{(\ell)}, t^{(\ell)}$.
        $c^\star \leftarrow c^{(L)}, \quad t \leftarrow \sum_\ell t^{(L)}, \quad \widehat{y} \leftarrow c^\star + t.$
        Compute losses: $\mathcal{L}_{\mathrm{pred}}$ (MSE), $\mathcal{L}_{\mathrm{corr}}$ (correlation), $\mathcal{L}_{\mathrm{mono}}$ (monotonicity), $\mathcal{L}_{\mathrm{triv}}$
        (residual).
        Total: $\mathcal{L}_{\mathrm{total}} = \mathcal{L}_{\mathrm{pred}} + \lambda_{\mathrm{caus}}\mathcal{L}_{\mathrm{corr}} + \lambda_{\mathrm{mono}}\mathcal{L}_{\mathrm{mono}} + \lambda_{\mathrm{unif}}\mathcal{L}_{\mathrm{triv}}$;
        optimizer step w.r.t. $\nabla \mathcal{L}_{\mathrm{total}}$.

---

0.494 to 0.358, with similar gains on FreeSolv and Lipo. This pattern aligns with Sec. 2.2. After node and edge splitting, the depthwise gate $w^{(\ell)}$ emphasizes the modality whose causal branch carries batch invariant evidence. The correlation schedule aligns each layer's causal scalar with the label, and the residual path learns remaining context with a stop gradient to prevent leakage into the causal branch. Fusion supplies complementary cues that the curriculum can promote and confounding that it can peel away, which strengthens accuracy.

In unimodal settings, the effect of causal peeling is limited and sometimes inconsistent. On *ESOL* and *Lipo*, both 2D and 3D variants exhibit only minor changes, and in some cases performance even degrades (e.g., 3D Lipo: $0.63 \to 0.64$ MAE). For *FreeSolv*, 2D and 3D improve slightly under causal splitting, but still remain far weaker than the multimodal fusion model. On *CycPeptMPDB*, 2D, 3D, and HELM all gain marginally, yet consistently trail behind the fused configuration.

## B.3 CROSS-MODAL CONSISTENCY

Table 6: Effect of consistency weight across datasets. Best per-dataset/metric in **Bold**.

| Setting | ESOL | | FreeSolv | | Lipo | | CycPeptMPDB | |
|---|---|---|---|---|---|---|---|---|
| | MAE | MSE | MAE | MSE | MAE | MSE | MAE | MSE |
| $\lambda_{\mathrm{cons}}$=0.5 | 0.5242 | 0.5106 | 1.0550 | 1.8375 | 0.5649 | 0.5002 | 0.3204 | 0.1934 |
| $\lambda_{\mathrm{cons}}$=1.0 | 0.5192 | 0.5050 | 1.1412 | 2.0513 | 0.5597 | 0.4949 | 0.3154 | 0.1839 |
| **CLaP** | **0.4456** | **0.3583** | **0.7020** | **0.8866** | **0.4672** | **0.3645** | **0.3056** | **0.1644** |

We vary a cosine-similarity penalty between modality-specific embeddings with weight $\lambda_{\mathrm{cons}}$ to test whether enforcing cross-modal agreement helps or hurts (Table 6). Across datasets, adding this penalty degrades the metrics compared to our default, with larger weights producing larger drops. The effect is most visible on FreeSolv and Lipo, and still present on CycPeptMPDB and ESOL.

This pattern is consistent with our design in Sec. 2.2. The model benefits from *complementarity* across modalities: the split routes label-relevant signal to the causal branch while the fusion gate selects the modality that carries the most batch-invariant evidence at each depth. Forcing embeddings to be too similar collapses modality-specific subspaces, reduces the gate's effective choice set, and makes batch-coupled shortcuts more likely to be shared. The correlation curriculum then has less diverse causal evidence to promote, and the residual path absorbs less of the confounding context. In practice, we keep $\lambda_{\mathrm{cons}}$ at zero or very small to preserve complementarity.

## B.4 PEELING DEPTH ON ADDITIONAL DATASETS

Table 7: Peeling depth ablation across three datasets. Best indicated in **Bold**.

| Layer | ESOL | | FreeSolv | | Lipo | |
|---|---|---|---|---|---|---|
| | MAE | MSE | MAE | MSE | MAE | MSE |
| 1 | 0.4879 | 0.4293 | 0.9733 | 1.9424 | 0.5201 | 0.4506 |
| **2** | **0.4456** | **0.3583** | **0.7020** | **0.8866** | 0.5079 | 0.4356 |
| 3 | 0.5495 | 0.5794 | 0.8987 | 1.4897 | **0.4672** | **0.3645** |
| 4 | 0.5463 | 0.5335 | 0.9849 | 1.5328 | 0.5376 | 0.4699 |

We extend the depth sweep to the remaining datasets. Table 7 shows a consistent U-shaped trend as the number of causal blocks increases. With too few layers the hierarchy *under-peels*. As depth grows beyond a dataset-dependent sweet spot the hierarchy *over-peels*.

ESOL and FreeSolv favor shallow hierarchies (two causal blocks), whereas Lipo benefits from one additional block, likely due to greater dataset complexity. This pattern is consistent with bias–variance tradeoffs and batch-wise invariance objective. The additional layer is useful when it tries to capture more complex signal.

In practice, we select depth by monitoring the layerwise correlation curve against the target schedule and choosing the smallest $L$ at which the validation error reaches its first stable minimum while correlations remain monotone and close to the anchors. Beyond that point additional layers yield diminishing returns and increase sensitivity to context.

## C ADDITIONAL CASE STUDIES AND MULTIMODALITY ANALYSIS

### C.1 ESOL DATASET

We benchmark on the ESOL dataset, which contains *1128 small organic molecules* with experimentally measured *aqueous solubility* reported as

$$\log S \equiv \log_{10}\big(\text{molar solubility in water (mol/L)}\big).$$

Thus, the sign directly encodes scale: positive values mean solubility $> 1$ mol/L, $\log S{=}0$ corresponds to $1$ mol/L, and negative values mean solubility $< 1$ mol/L.

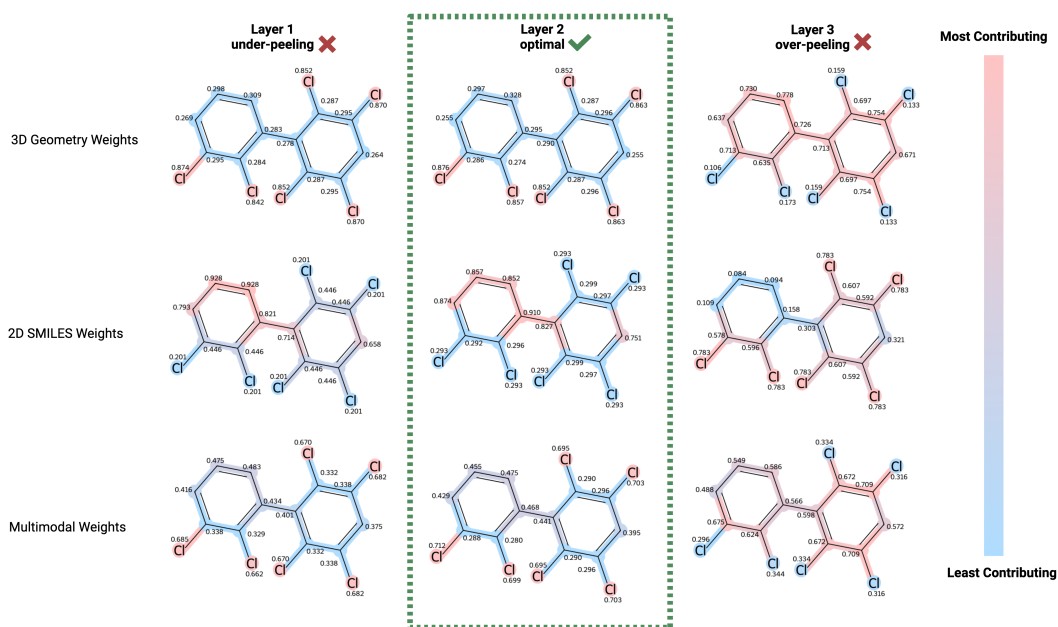

Figure 6: Layer-wise attribution maps across input modalities. Red colored atoms are attributed with affecting the molecule's water affinity the most.

At the optimal depth (L2), CLaP assigns the highest weights to chloro substituents and adjacent aryl carbons that cluster via short Cl–C/Cl–Cl distances in 3D; the fused readout preserves this through-space signal while the 2D branch highlights halogenated positions from connectivity alone. This depth-wise focus is exactly the intended "peel": batch-coupled context is removed, and label-relevant structure (a single hydrophobic patch) is retained. The outcome matches physical chemistry: contiguous chlorination increases non-polar surface and reduces hydrogen-bonding capacity, driving logS lower. Under-peeling spreads attribution across the ring (shortcuts), and over-peeling reintroduces residual context, illustrating why CLaP's layerwise schedule matters.

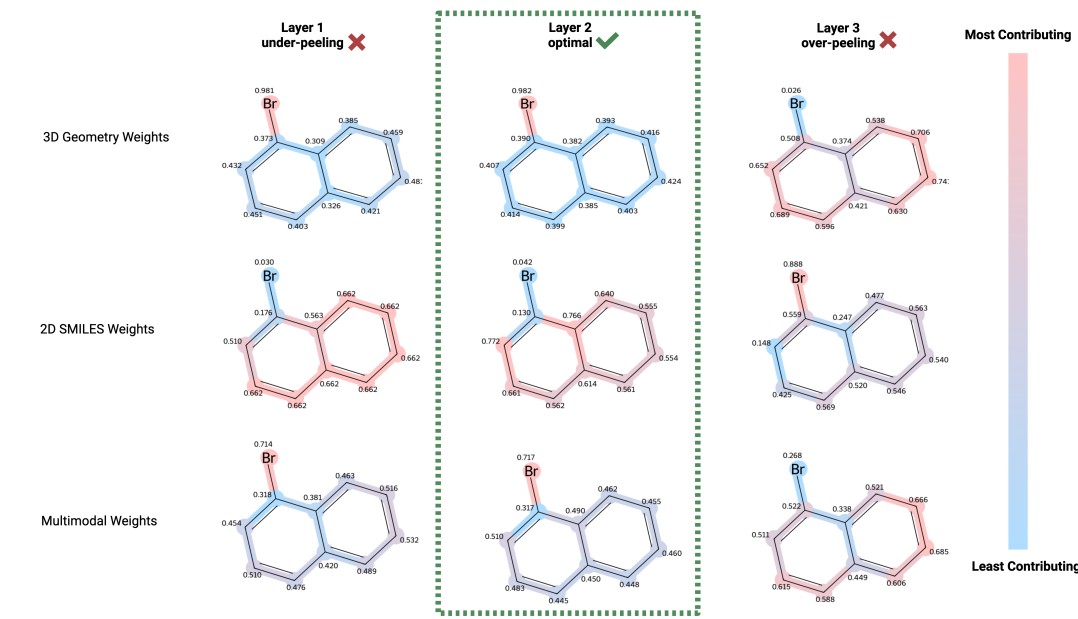

Figure 7: Layer-wise attribution maps across input modalities. Red colored atoms are attributed with affecting the molecule's insolubility the most.

At the optimal depth (L2), CLaP assigns the highest weights to the *aryl bromide* and the *fused bicyclic ring junction*; the 3D branch aggregates the planar $\pi$-surface into a single hydrophobic patch, while the 2D branch captures bromination and ring fusion from connectivity alone. This depth-wise focus is exactly the intended "peel": batch-coupled context is removed, and label-relevant structure (a large, contiguous hydrophobic surface) is retained. The outcome matches physical chemistry: expanded aryl surface together with Br increases lipophilicity and contributes essentially no hydrogen-bond donors/acceptors, yielding minimal polar counterbalance and driving $\log S$ more negative (lower aqueous solubility). Under-peeling spreads attribution across the scaffold (shortcuts), and over-peeling smears weights and reintroduces residual context. CLaP effectively captures a general rule in physical chemistry: addition of halogen groups increases the molecule's overall solubility in organic solvents and decreases the molecule's overall solubility in aqueous solvents.

## C.2    FREESOLV DATASET

We benchmark on the FreeSolv dataset, which contains *642 small organic molecules* with experimentally measured *hydration free energies in water* reported as

$$\Delta G_{\text{hyd}} \text{ [kcal/mol]}.$$

Thus, the sign directly encodes favorability: *negative* values mean exergonic (favorable) hydration and greater water affinity, values near 0 indicate weak preference, and *positive* values mean endergonic (unfavorable) hydration and greater hydrophobicity.

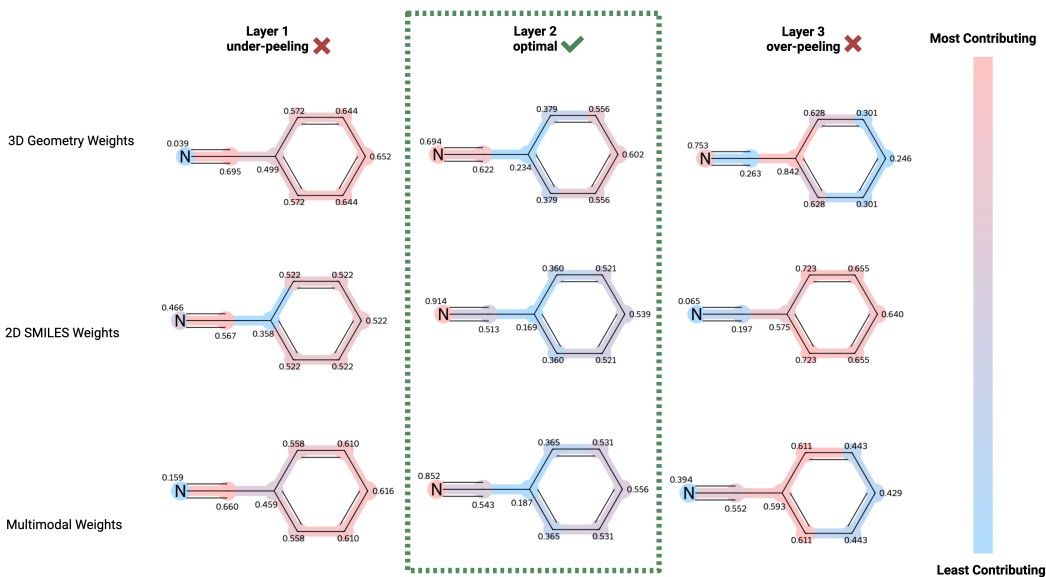

Figure 8: Layer-wise attribution maps across input modalities. Red colored atoms are attributed with affecting the molecule's hydrophilicity the most.

At the optimal depth (L2), CLaP assigns the highest weights to the *nitrile substituent* and the adjacent aromatic carbons; the 3D branch aggregates the linear C≡N dipole into a localized polar feature, while the 2D branch captures the nitrile group and ring context from connectivity. This depth-wise focus is exactly the intended "peel": batch-coupled context is removed, and label-relevant structure (a polar handle appended to an aryl surface) is retained. The outcome matches physical chemistry: the nitrile increases polarity and provides a strong dipole acceptor, raising aqueous solubility relative to an unsubstituted aryl. Nitriles are widely used in medicinal chemistry contexts as metabolically stable polar groups that tune solubility and binding while maintaining a compact scaffold.

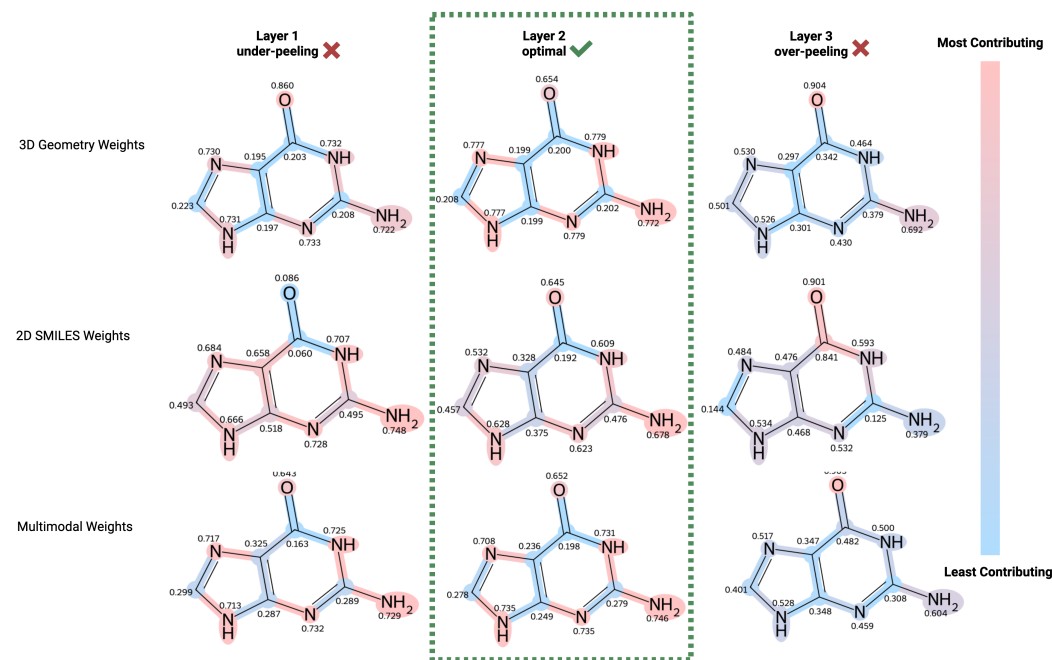

Figure 9: Layer-wise attribution maps across input modalities. Red colored atoms are attributed with affecting the molecule's hydrophilicity the most.

At the optimal depth (L2), CLaP assigns the highest weights to the *carbonyl oxygen* and the *exocyclic amine group*, with additional contributions from the ring nitrogens. The 3D branch aggregates the localized dipoles into a polar spine, while the 2D branch captures the carbonyl and amino substituents from connectivity. This matches physical chemistry: the carbonyl oxygen and multiple $NH_2$/NH groups provide hydrogen-bond acceptors and donors, substantially raising aqueous solubility relative to a purely carbocyclic core. Under-peeling diffuses weights across the ring scaffold (shortcuts), while over-peeling smears attribution and reintroduces residual context. Heteroaryl amides of this type are commonly found in nucleobases and drug-like scaffolds, where their balance of hydrogen-bonding capacity and planarity helps tune solubility in chemical biology.

## C.3 LIPO DATASET

We benchmark on the Lipophilicity (Lipo) dataset which contains *4,200 small molecules* curated experimentally. The target is the octanol/water distribution coefficient at physiological pH ($\log D_{7.4}$), defined as

$$\log D_{7.4} \;\equiv\; \log_{10}\left(\frac{[C]_{\text{octanol}}}{[C]_{\text{water}}}\right)_{\text{pH}=7.4},$$

where concentrations include all ionization states at pH 7.4. Thus, *higher (more positive)* $\log D_{7.4}$ denotes *greater lipophilicity* (preference for octanol), whereas *lower/negative* values indicate *greater hydrophilicity*.

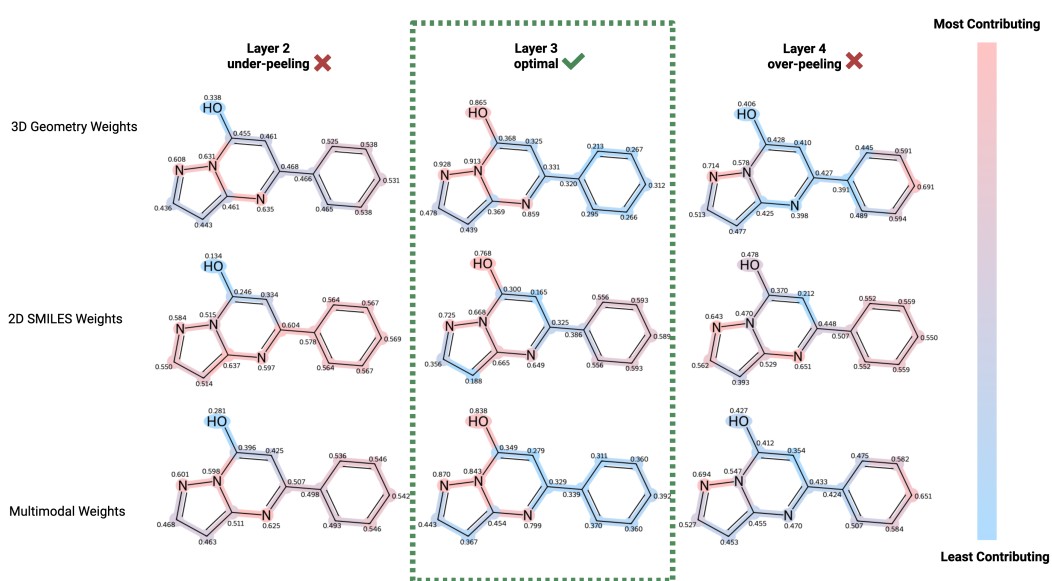

Figure 10: Layer-wise attribution maps across input modalities. Red colored atoms are attributed with affecting the molecule's hydrophilicity the most.

At the optimal depth (L3), CLaP assigns the highest weights to the *hydroxyl substituent* and nearby heteroaryl nitrogens. The 3D branch aggregates the O–H dipole into a localized polar hotspot, while the 2D branch highlights the hydroxyl group and its attachment point to the heteroaromatic ring. This depth-wise focus is exactly the intended "peel": batch-coupled context is removed, and label-relevant structure (polar donors/acceptors distributed across a largely aromatic scaffold) is retained. The outcome matches physical chemistry: the hydroxyl provides a strong hydrogen-bond donor and acceptor site, partially counterbalancing the extended aromatic surface. As a result, the molecule is predicted to be more *hydrophilic* than a fully unsubstituted aromatic analog.

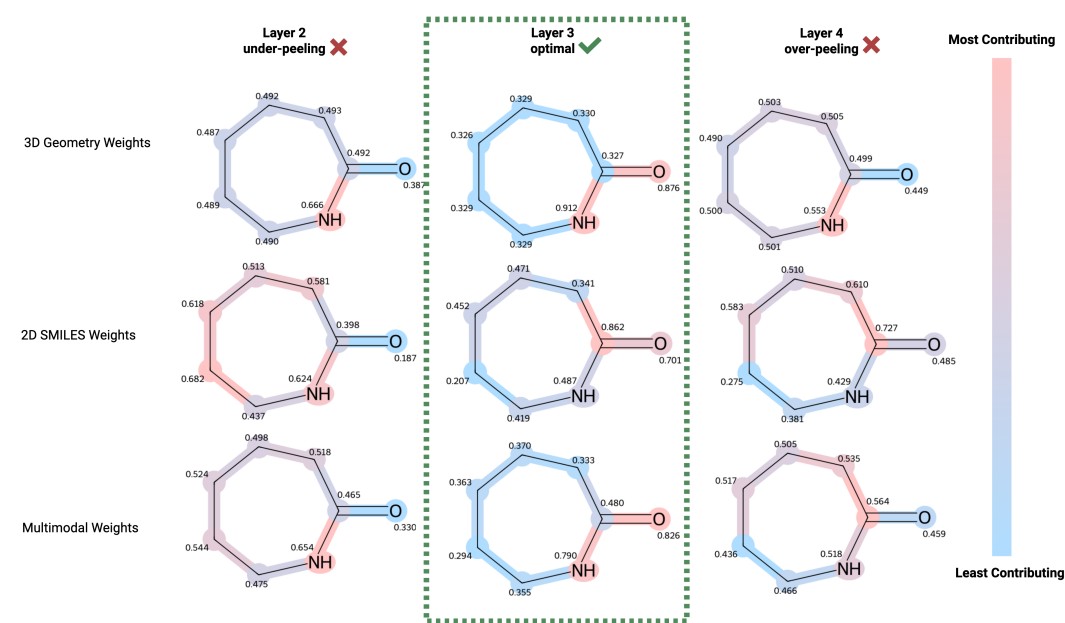

Figure 11: Layer-wise attribution maps across input modalities. Red colored atoms are attributed with affecting the molecule's hydrophilicity the most.

At the optimal depth (L3), CLaP assigns the highest weight to the *amide carbonyl oxygen* and the nearby amide nitrogen. The 3D branch captures the strong local dipole between the C=O and NH, while the 2D branch highlights the amide linkage from connectivity. This depth-wise focus is exactly the intended "peel": batch-coupled context is removed, and label-relevant structure (a polar amide motif appended to a hydrophobic ring) is retained. CLaP captures a general modification in chemistry: the amide group generally introduces both a hydrogen-bond donor and acceptor, increasing polarity and reducing lipophilicity relative to an unsubstituted carbocyclic scaffold. As a result, the molecule is predicted to be more *hydrophilic* (lower $\log D_{7.4}$).

## C.4 CYCPEPTMPDB DATASET

We benchmark on the CycPeptMPDB dataset, which contains *7,334 cyclic peptides* with experimentally measured *passive membrane permeability* reported on a logarithmic scale as

$$\log P \;\equiv\; \log_{10}\!\big(P\,[\text{cm/s}]\big).$$

Thus, higher (less negative) $\log P$ denotes more permeable peptides, whereas more negative values indicate poor permeability.

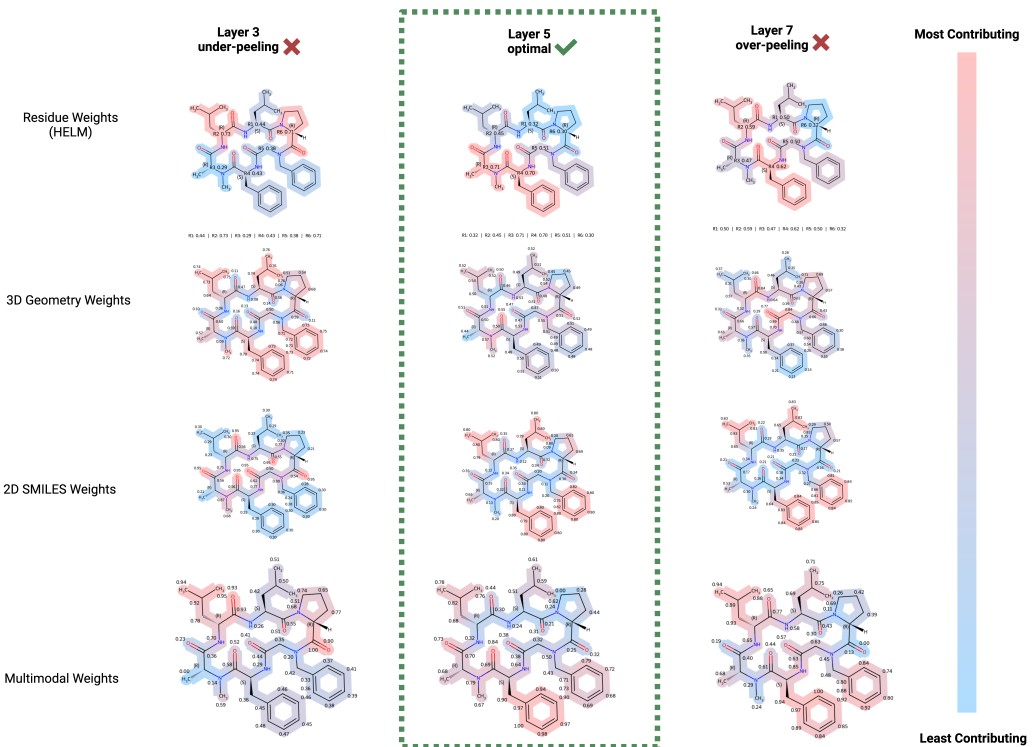

Figure 12: Layer-wise attribution maps across input modalities. Red colored atoms are attributed with affecting the molecule's permeability the most.

At the optimal depth (L5), CLaP highlights *hydrophobic residues* such as phenylalanine and aliphatic side chains as the dominant contributors. The 3D branch aggregates bulky aromatic surfaces into hydrophobic patches, while the HELM representation captures residue-level context. This depth-wise focus is exactly the intended "peel": batch-coupled noise is removed, and label-relevant features (hydrophobic residues that enhance membrane passage) are retained. The outcome is consistent with physical chemistry and with what is known for peptide therapeutics: aromatic and aliphatic substitutions increase lipophilicity and improve passive *cell permeability*. More broadly, such analyses underscore the drug-design logic of cyclic and linear peptide scaffolds: by balancing polarity (for solubility) with nonpolar residues (for permeability), peptide-based molecules can be optimized as potential drugs. Peptide drugs already play a major role in therapy, ranging from hormones and enzyme inhibitors to next-generation modulators of protein–protein interactions.

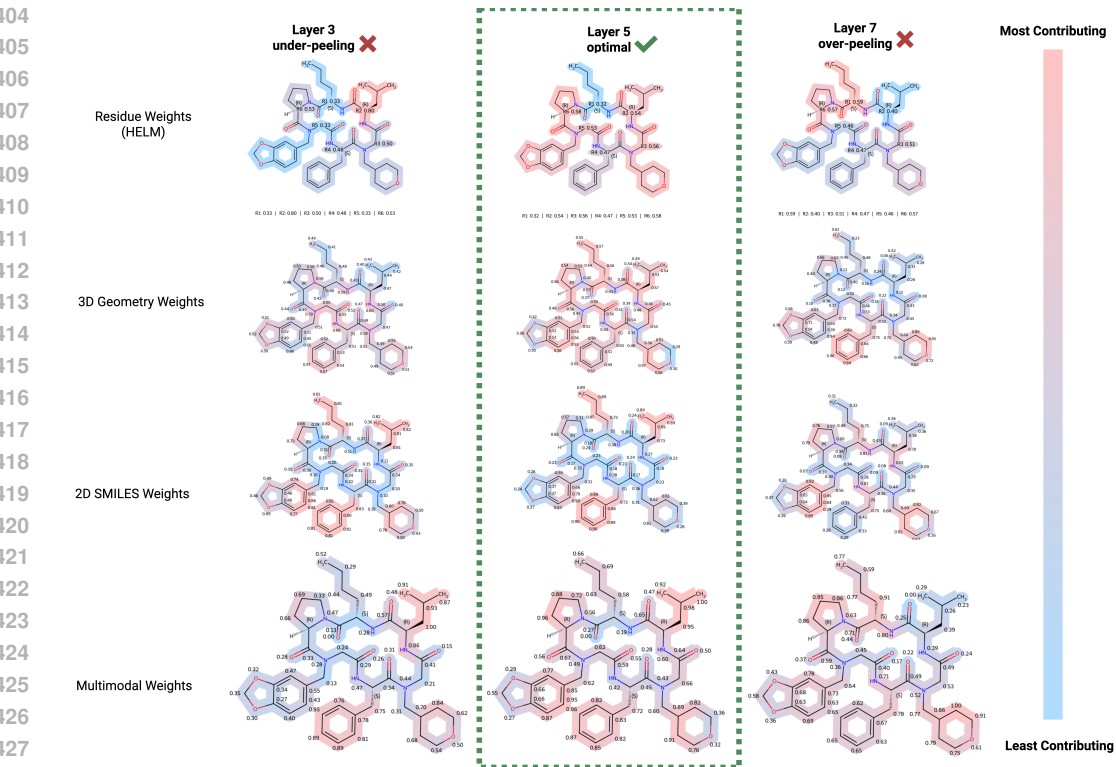

Figure 13: Layer-wise attribution maps across input modalities. Red colored atoms are attributed with affecting the molecule's permeability the most.

At the optimal depth (L5), CLaP again highlights *hydrophobic residues* such as aromatics and aliphatic side chains as the main drivers of permeability. These groups form contiguous lipophilic patches, favoring bilayer partitioning and consistent with design principles for permeable peptides. In contrast, *heteroatom-rich sites* such as oxygen incorporated cycles contribute less, reflecting their polarity and the desolvation penalty that reduces passive diffusion. Thus, the model recovers a chemically intuitive balance: hydrophobic groups increase permeability, while oxygen and general incorporation of polar atoms in the side chains reduce permeability. CLaP is able to distinguish both drivers of permeability, as well as identify motifs that harm permeability, suggesting molecular edits that chemists can make to optimize the peptide.

# D  IMPLEMENTATION DETAILS

## D.1  PROPOSED FRAMEWORK IMPLEMENTATION DETAILS

For *CycPeptMPDB*, we encode 2D molecular graphs and HELM strings with GAT and 3D geometries with EGNN; each backbone has 3 layers, producing 128-D embeddings that are linearly projected to a shared 256-D space. The target correlation schedule increases linearly from $0.5$ to $0.8$ across $L{=}5$ causal blocks. Training runs for 100 epochs with batch size 16 on $8\times$ RTX-6000 GPUs with early stopping. Loss weights are $\lambda_{\mathrm{caus}}{=}1.0$, $\lambda_{\mathrm{unif}}{=}1.0$, and $\lambda_{\mathrm{mono}}{=}0.5$; cross-modal consistency is disabled ($\lambda_{\mathrm{cons}}{=}0$), and the hinge margin is set to $0$. For *ESOL*, *FreeSolv*, and *Lipo* (non-peptide datasets), we use two modalities (2D graphs and 3D geometries); unless noted otherwise, embeddings are projected to 256-D and the correlation schedule is $0.4{\rightarrow}0.7$. ESOL uses width 128 with 2 encoder layers and $L{=}2$ causal blocks; FreeSolv uses width 64 with 2 layers and $L{=}2$; Lipo uses width 64 with 3 layers and $L{=}3$. The hinge margin is $0$ for all settings.

## D.2  BASELINES IMPLEMENTATION DETAILS

For causality-oriented baselines, we modified the original classification-based causal framework into regression models for molecular property prediction. Specifically, the classification head was replaced with a regression head by fixing the output layer dimension to one. At the data level, we replaced the original discrete datasets with SMILES-derived PyG graph structures labeled with continuous molecular properties (e.g., permeability, logD74). For evaluation, metrics were changed from accuracy/AUC to MSE, RMSE, MAE, and R². These modifications preserve the structural features of the original architecture while adapting it to our new regression tasks.

# E  THE USE OF LARGE LANGUAGE MODELS (LLMS)

We used an LLM solely for light editorial assistance (grammar, wording, and minor style). The LLM did not contribute ideas, methods, models, experiments, analyses, or decisions, and was not used to generate or label data or code. All scientific content is by the authors.

