# OpenReview forum: "Peeling Context from Cause for Multimodal Molecular Property Prediction"
_ICLR.cc/2026/Conference — Submitted to ICLR 2026_

### Official Review · Reviewer_iB76 · 2025-10-28

**Soundness:** 2
**Presentation:** 2
**Contribution:** 2
**Rating:** 4
**Confidence:** 4

**Summary:**

The paper tackles spurious “context” in molecular property prediction and proposes CLaP, a layerwise framework to extract causal information from irrelevant contexts. At each depth, representations are split into a causal branch and a trivial (context) branch, and the causal path is trained to satisfy an increasing target for within-batch label correlation with a monotonicity regularizer. The trivial path fits the residuals. Multimodal fusion (2D graphs, 3D geometry, peptide strings) occurs on the causal path, yielding atom-level saliency. On ESOL, FreeSolv, Lipo, and CycPeptMPDB, CLaP improves the regression error compared to the baselines; ablations confirm each component’s contribution, and re-batching tests suggest robustness to batch artifacts.

**Strengths:**

- The authors successfully implement the idea of separating useful information from irrelevant background.

- By enforcing a gradually increasing correlation coefficient between intermediate-layer representations and the label, which enabled informative saliency maps.

**Weaknesses:**

- The empirical performance needs to be more convincing: the compared baseline models are largely outdated (pre-2022), and head-to-head evaluations against more recent methods [3][4] are necessary to validate the claimed gains. Moreover, reporting results on more tasks such as QM9 [1] and BACE [2], among others, would provide a more comprehensive assessment.

- The observed improvements may stem primarily from the introduction of multimodality rather than from the proposed causal-context peeling itself, and this confound should be disentangled.

- Since the proposed approach relies on optimizing the correlation coefficient with continuous label values, it appears to be difficult to apply directly to classification task settings.

**Questions:**

- Since Equation (7) already enforces deeper layers to achieve higher correlation with the ground-truth labels, isn’t $L_{mono}$ a redundant training objective? How do you explain the substantial performance gap observed in Table 4 when $L_{mono}$ is included versus omitted?

- If the trivial branch is intended to capture information deemed irrelevant and therefore “peeled” away, why are these trivial signals accumulated and reused in the final prediction? What theoretical or empirical justification supports re-introducing them at the output?

---

> ### Author Response · Authors · 2025-11-21
>
> We truly appreciate your time and thoughtful comments, as well as the helpful and constructive feedback! Here, we give point-by-point responses to your comments.(Please note that we have uploaded the revised PDF)
>
> **Weakness**
>
> >**W1: The evaluation relies on outdated baselines and too few tasks; newer models and more benchmarks are needed to convincingly validate the performance gains.**
>
> Thank you for the suggestions. We have now incorporated scaffold-based OOD splits on five additional TDC datasets, **Half_Life_Obach, Solubility, LD50_Zhu, Hepatocyte, and Microsome**  covering absorption, excretion, and toxicity. We also added more latest baselines [1,2,3], including **CGR**, one of the latest causality-oriented models, to provide a more comprehensive comparison.
>
> Except for Solubility and Lipo (where performance are already near the best), our model outperforms all baselines on the remaining datasets. These results are included in the revised version.
>
> **Comparison across nine benchmarks**
> (Best per-dataset MAE in bold)
>
> | Method | ESOL | FreeSolv | Lipo | CycPept | HalfLife | Hepatocyte | Microsome | Solubility | Toxicity |
> |--------|-------|-----------|--------|-----------|-------------|-------------|-------------|-------------|-----------|
> | CAL | 0.4923 | 1.1273 | 0.6217 | 0.3682 | 1.0189 | 0.9839 | 0.8335 | 1.0572 | 0.6636 |
> | FP-GNN | 0.5039 | 1.1561 | 0.4809 | 0.3487 | 0.8857 | 1.0836 | 0.8083 | 0.8383 | 0.5242 |
> | CGR | 1.5853 | 2.7268 | 0.9670 | 0.5792 | 1.0360 | 1.2371 | 1.2544 | 1.6741 | 0.7794 |
> | GSL | 1.0032 | 2.3476 | 0.9007 | 0.6590 | 0.9365 | 1.1045 | 1.0723 | 1.5883 | 0.7414 |
> | DIR | 0.6267 | 2.2420 | 0.6328 | 0.4243 | 0.9600 | 1.0780 | 1.1580 | 1.1890 | 0.6810 |
> | MAT | 0.5394 | 0.7680 | 0.5284 | 0.4129 | 0.9915 | 1.5660 | 1.1350 | 0.9250 | 0.8622 |
> | MolFCL | 0.4660 | 1.0300 | **0.4330** | 0.3690 | 0.9210 | 1.1160 | 0.9430 | **0.6990** | 0.7530 |
> | **CLaP** | **0.4456** | **0.7020** | 0.4672 | **0.3056** | **0.7832** | **0.9520** | **0.7221** | 0.7725 | **0.5064** |
>
>
> We appreciate your recommended datasets and baselines and will integrate them as well.
> We hope this addresses your concerns regarding model evaluation.
>
>
> \[1\] Xiang Tang, Qichang Zhao, Jianxin Wang, Guihua Duan.
> *MolFCL: predicting molecular properties through chemistry-guided contrastive and prompt learning.*
> Bioinformatics, 41(2), 2025.
>
> \[2\] Bangyi Zhao, Weixia Xu, Jihong Guan, Shuigeng Zhou.
> *Molecular property prediction based on graph structure learning.*
> Bioinformatics, 40(5), 2024.
>
> \[3\] Yujia Yin, Tianyi Qu, Zihao Wang, Yifan Chen.
> *A Recipe for Causal Graph Regression: Confounding Effects Revisited.*
> ICML 2025.

---

> > ### Author Response · Authors · 2025-11-21
> >
> > >**W2: The gains may come from using multimodal inputs rather than the causal peeling method itself, and this needs to be disentangled.**
> >
> > Thank you for this insightful comment. To disentangle the effect of multimodality from the causal–trivial peeling itself, we conducted a targeted causal ablation study (Table 1). Under identical multimodal configurations, models **with** CLaP consistently outperform those **without** it across all nine datasets, confirming that the gains are not solely due to multimodality.
> >
> > **Comparison: w/o causal split vs. CLaP**
> > (Best per-dataset MAE in bold)
> >
> > | Method | ESOL | FreeSolv | Lipo | CycPept | HalfLife | Hepatocyte | Microsome | Solubility | Toxicity |
> > |--------|-------|-----------|--------|-----------|-------------|-------------|-------------|-------------|-----------|
> > | w/o causal split | 0.5094 | 1.2858 | 0.5145 | 0.3120 | 0.8301 | 1.0571 | 0.7908 | 0.7850 | 0.5184 |
> > | **CLaP** | **0.4456** | **0.7020** | **0.4672** | **0.3056** | **0.7832** | **0.9520** | **0.7221** | **0.7725** | **0.5064** |
> >
> >
> >
> >
> > Furthermore, as shown in Appendix B.2, applying CLaP to a single modality yields limited improvements. As causal splitting leverages complementary causal information across modalities, allowing the model to fully exploit causal information that may not be present in any single view.
> >
> > **Effect of causal peeling under *unimodal* vs. *multimodal* settings**
> > (Best per-dataset/metric in bold)
> >
> > | Setting                  | ESOL MAE | ESOL MSE | FreeSolv MAE | FreeSolv MSE | Lipo MAE | Lipo MSE | CycPeptMPDB MAE | CycPeptMPDB MSE |
> > |--------------------------|---------:|---------:|-------------:|-------------:|---------:|---------:|----------------:|----------------:|
> > | **Multimodal (w split)** | **0.4456** | **0.3583** | **0.7020** | **0.8866** | **0.4672** | **0.3645** | **0.3056** | **0.1644** |
> > | Multimodal (w/o split)   | 0.5094 | 0.4939 | 1.2858 | 2.5719 | 0.5145 | 0.4302 | 0.3120 | 0.1834 |
> > | 2D Graph (w split)       | 0.9453 | 1.5993 | 1.1267 | 1.9876 | 0.5171 | 0.4506 | 0.4068 | 0.2889 |
> > | 2D Graph (w/o split)     | 0.9631 | 1.6931 | 1.2003 | 2.4157 | 0.5230 | 0.4504 | 0.4077 | 0.2884 |
> > | 3D Geometry (w split)    | 0.5322 | 0.5235 | 1.0640 | 1.8055 | 0.6412 | 0.6519 | 0.4221 | 0.3102 |
> > | 3D Geometry (w/o split)  | 0.5219 | 0.5036 | 1.0721 | 1.9262 | 0.6260 | 0.6449 | 0.4244 | 0.3071 |
> > | HELM (w split)           | -- | -- | -- | -- | -- | -- | 0.3310 | 0.2023 |
> > | HELM (w/o split)         | -- | -- | -- | -- | -- | -- | 0.3194 | 0.1910 |
> >
> >
> > >**W3: Since the proposed approach relies on optimizing the correlation coefficient with continuous label values, it appears to be difficult to apply directly to classification task settings.**
> >
> >
> > Thank you for this insightful observation. While this paper focuses on molecular property prediction as a regression task, the proposed CLaP framework is not limited to regression. The correlation-based causal alignment can be naturally extended to classification, as noted in the discussion section(Section 5). In such a setup, each layer’s output can connect to a classification head while keeping the causal–trivial split unchanged, since the framework only introduces gating for causal splitting and leaves the core architecture and learning mechanism intact.
> >
> >
> > **Questions**
> >
> > >**Q1:If deeper layers are already optimized to align with labels (Eq. 7), why is $L_{\text{mono}}$ still needed, and how does it cause such a large performance difference?**
> >
> > Thank you for the question. Equation (7) alone does not provide sufficiently strong or distinctive supervision across layers. Our target correlation schedule ranges from 0.4 to 0.7, and when many layers are involved, the correlation gap between adjacent layers becomes very small. This weakens the depth-wise training signal and makes it vulnerable to batch-dependent noise.
> >
> > As a result, relying only on Eq. (7) can lead to ambiguous layer ordering and unstable causal–trivial separation. The monotonicity constraint in Eq. (9) is therefore essential rather than redundant — it enforces a strictly increasing correlation pattern, ensuring consistent causal refinement across layers. We hope this clarifies the necessity of Eq. (9).
> >
> >
> > We hope this demonstrates how the framework can be generalized to classification.

---

> > > ### Author Response · Authors · 2025-11-21
> > >
> > > >**Q2:If trivial signals are meant to be removed, why are they added back at the end, and what justifies reintroducing them in the final prediction?**
> > >
> > > Thank you for the question. The branch named “trivial” is more accurately described as a *context branch* (we have updated the terminology in the revised version). Its purpose is to explicitly model residual, non-causal components including batch-dependent or unstable environmental factors such as scaffold biases introduced by batch composition.
> > >
> > > Although these factors are not causally meaningful, they can still have some consistent and measurable influence on prediction. For this reason, we reintroduce the context signal at the end, allowing the model to benefit from these residual patterns while keeping the causal pathway clean, stable, and interpretable. A similar treatment of confounders can be found in \[1\].
> > >
> > > As shown in Appendix A.2, the context branch reliably learns the expected residual component, providing necessary calibration for accurate prediction without contaminating the causal representation.
> > >
> > >
> > > \[1\] Yujia Yin, Tianyi Qu, Zihao Wang, Yifan Chen.
> > > *A Recipe for Causal Graph Regression: Confounding Effects Revisited.*
> > > ICML 2025.
> > >
> > >
> > > **We appreciate the reviewers’ insights and hope our detailed responses have addressed the concerns.**

---

### Official Review · Reviewer_Trmx · 2025-10-30

**Soundness:** 3
**Presentation:** 3
**Contribution:** 2
**Rating:** 4
**Confidence:** 4

**Summary:**

This paper proposes Causal Layerwise Peeling, a framework for multimodal molecular property prediction that explicitly separates causal signals from contextual noise in a layerwise manner. CLaP introduces: 1) a causal–trivial split at each layer, 2) a batch-wise invariance principle to isolate sample-intrinsic signals, 3) a depth-wise correlation schedule and monotonicity regularizer, and 4) multimodal fusion across 2D SMILES graphs, 3D geometries, and peptide notation. Empirical results on four molecular benchmarks show consistent gains over strong baselines. The model also provides atom-level causal saliency maps aligned with chemical intuition.

**Strengths:**

1. The proposed method effectively distinguishes causal and contextual information structurally, overcoming limitations of traditional single-stage causal subgraph selection methods. This approach is conceptually novel and offers strong interpretability. The authors also provide formal derivations and theoretical proofs, which elucidate the model's rationale for progressively removing contextual noise.
2. The ablation studies are thorough, systematically validating the impact of key modules (causal-trivial branch, correlation scheduling, monotonicity regularization). This forms a logically complete validation loop.
3. The work introduces multimodal fusion into causal representation learning. By adaptively gating and fusing 2D, 3D, and sequential modalities, the model enhances robustness and cross-modal consistency.

**Weaknesses:**

1. The comparative experiments are not sufficiently comprehensive. The authors have only evaluated on four datasets, which are limited in size. We recommend augmenting the comparison with experiments on the GOOD benchmark. Additionally, the causality-oriented baselines are only compared against OOD methods, without inclusion of molecular causality-oriented baselines[1]. Please include relevant comparative experiments.
2. Causal invariance is validated solely through "re-batching" experiments. This needs to be compared with other OOD settings to further justify the chosen approach. The assumption of independent and identically distributed mini-batch sampling may not hold in practice for chemical data, where clustering or scaffold biases are common. Such biases may not adequately represent real-world distribution shifts (e.g., scaffold splits or property distribution shifts).
3. Although the paper claims performance improvements from multimodal fusion, it lacks an analysis of robustness under conditions where one or more modalities are missing or noisy.

Ref:

[1] Learning Substructure Invariance for Out-of-Distribution Molecular

**Questions:**

1. Given the limited number of benchmarks used in this work, what are the training costs and convergence stability of the model when trained on larger datasets or more complex molecular systems?
2. If all modalities contain significant batch-dependent noise, will the proposed causal-trivial partition still converge effectively?
3. Could you report results on OOD settings (e.g., scaffold-based splits) to further validate the causal generalization capabilities of your model?

---

> ### Author Response · Authors · 2025-11-21
>
> Thank you for your time and for providing such thoughtful and constructive feedback. Below, we respond to each of your comments in turn.(Please note that we have uploaded the revised PDF)
>
> **Weakness**
>
> >**W1 & 2 &Q3: The experiments are limited and miss key baselines, especially molecular causality methods and OOD benchmark comparisons.**
>
> Thank you for the suggestions. We adopted scaffold-based OOD splits on five additional TDC datasets — **Half_Life_Obach, Solubility, LD50_Zhu, Hepatocyte, and Microsome** — covering absorption, excretion, and toxicity properties. We also include additional baselines \[1, 2, 3\], including **CGR**, one of the latest causality-oriented molecular models.
>
> **Comparison across nine benchmarks**
> (Best per-dataset MAE in bold)
>
> | Method | ESOL | FreeSolv | Lipo | CycPept | HalfLife | Hepatocyte | Microsome | Solubility | Toxicity |
> |--------|-------|-----------|--------|-----------|-------------|-------------|-------------|-------------|-----------|
> | CAL | 0.4923 | 1.1273 | 0.6217 | 0.3682 | 1.0189 | 0.9839 | 0.8335 | 1.0572 | 0.6636 |
> | FP-GNN | 0.5039 | 1.1561 | 0.4809 | 0.3487 | 0.8857 | 1.0836 | 0.8083 | 0.8383 | 0.5242 |
> | CGR | 1.5853 | 2.7268 | 0.9670 | 0.5792 | 1.0360 | 1.2371 | 1.2544 | 1.6741 | 0.7794 |
> | GSL | 1.0032 | 2.3476 | 0.9007 | 0.6590 | 0.9365 | 1.1045 | 1.0723 | 1.5883 | 0.7414 |
> | DIR | 0.6267 | 2.2420 | 0.6328 | 0.4243 | 0.9600 | 1.0780 | 1.1580 | 1.1890 | 0.6810 |
> | MAT | 0.5394 | 0.7680 | 0.5284 | 0.4129 | 0.9915 | 1.5660 | 1.1350 | 0.9250 | 0.8622 |
> | MolFCL | 0.4660 | 1.0300 | **0.4330** | 0.3690 | 0.9210 | 1.1160 | 0.9430 | **0.6990** | 0.7530 |
> | **CLaP** | **0.4456** | **0.7020** | 0.4672 | **0.3056** | **0.7832** | **0.9520** | **0.7221** | 0.7725 | **0.5064** |
>
>
> Except for the Solubility and Lipo dataset, where performance are already near the top, our model outperforms all baselines across the remaining datasets. These updated results are included in the revised version. We also appreciate the additional baseline you suggested and will include it as well.
>
> We hope this addresses your concern.
>
> \[1\] Xiang Tang, Qichang Zhao, Jianxin Wang, Guihua Duan.
> *MolFCL: predicting molecular properties through chemistry-guided contrastive and prompt learning.*
> Bioinformatics, 41(2), 2025. https://doi.org/10.1093/bioinformatics/btaf061
>
> \[2\] Bangyi Zhao, Weixia Xu, Jihong Guan, Shuigeng Zhou.
> *Molecular property prediction based on graph structure learning.*
> Bioinformatics, 40(5), 2024. https://doi.org/10.1093/bioinformatics/btae304
>
> \[3\] Yujia Yin, Tianyi Qu, Zihao Wang, Yifan Chen.
> *A Recipe for Causal Graph Regression: Confounding Effects Revisited.*
> ICML 2025.
>
>
> >**W3: The paper shows multimodal gains but does not test robustness when modalities are missing or noisy.**
>
> Thanks for pointing this out. In **Table 5**, we provide both multimodal and unimodal results. In the unimodal case, applying causal peeling can introduce instability due to limited single-modality information. However, in the multimodal setting, performance improves, as causal splitting leverages complementary causal information across modalities, allowing the model to fully exploit causal information that may not be present in any single view.
>
> **Effect of causal peeling under *unimodal* vs. *multimodal* settings**
> (Best per-dataset/metric in bold)
>
> | Setting                  | ESOL MAE | ESOL MSE | FreeSolv MAE | FreeSolv MSE | Lipo MAE | Lipo MSE | CycPeptMPDB MAE | CycPeptMPDB MSE |
> |--------------------------|---------:|---------:|-------------:|-------------:|---------:|---------:|----------------:|----------------:|
> | **Multimodal (w split)** | **0.4456** | **0.3583** | **0.7020** | **0.8866** | **0.4672** | **0.3645** | **0.3056** | **0.1644** |
> | Multimodal (w/o split)   | 0.5094 | 0.4939 | 1.2858 | 2.5719 | 0.5145 | 0.4302 | 0.3120 | 0.1834 |
> | 2D Graph (w split)       | 0.9453 | 1.5993 | 1.1267 | 1.9876 | 0.5171 | 0.4506 | 0.4068 | 0.2889 |
> | 2D Graph (w/o split)     | 0.9631 | 1.6931 | 1.2003 | 2.4157 | 0.5230 | 0.4504 | 0.4077 | 0.2884 |
> | 3D Geometry (w split)    | 0.5322 | 0.5235 | 1.0640 | 1.8055 | 0.6412 | 0.6519 | 0.4221 | 0.3102 |
> | 3D Geometry (w/o split)  | 0.5219 | 0.5036 | 1.0721 | 1.9262 | 0.6260 | 0.6449 | 0.4244 | 0.3071 |
> | HELM (w split)           | -- | -- | -- | -- | -- | -- | 0.3310 | 0.2023 |
> | HELM (w/o split)         | -- | -- | -- | -- | -- | -- | 0.3194 | 0.1910 |
>
> This aligns with our expectation that causal–trivial decomposition is most effective when multiple modalities contribute distinct causal information. We hope this clarifies the observed behavior.

---

> > ### Author Response · Authors · 2025-11-21
> >
> > **Questions**
> >
> > >**Q1:How well does the model scale in training cost and stability when applied to larger or more complex datasets?**
> >
> > Thank you for the question. While our current benchmarks are moderate in size, the proposed framework is computationally lightweight. CLaP introduces only a simple layerwise split and two scalar objectives (correlation and monotonicity), without adding extra encoders or modality-specific networks.
> >
> > In practice, the training cost remains comparable to a standard GNN or transformer of similar depth, and we observe stable convergence across all datasets.
> >
> >
> > >**Q2: If all modalities contain significant batch-dependent noise, will the proposed causal-trivial partition still converge effectively?**
> >
> > Thank you for the question. Our method does not require any modality to be “clean.” Instead, it assumes only that some degree of cross-batch stable structure exists across modalities, an assumption that typically holds in molecular property prediction.
> >
> > Since molecular properties are fundamentally driven by atomic-level chemical patterns, even if one modality is noisy, they still share underlying structure that allows the model to capture atom-level causal signals.
> >
> > As with any model, data quality influences how clearly stable components can be extracted. However, the causal–trivial split helps by isolating batch-dependent noise into the trivial branch.
> >
> > Our experiments support this: CycPeptMPDB, the noisiest dataset (SMILES lacks full peptide structural info), still shows that multimodal CLaP achieves the best and most stable performance. This demonstrates that even when all modalities contain noise, CLaP can extract stable causal signals while absorbing noise into the trivial path.
> >
> >
> > **Thank you for taking the time to review our work. We hope the explanations above resolve the concerns raised and give a clearer picture of our contribution.**

---

### Official Review · Reviewer_3uWx · 2025-10-31

**Soundness:** 3
**Presentation:** 2
**Contribution:** 2
**Rating:** 4
**Confidence:** 4

**Summary:**

The paper introduces CLaP (Causal Layerwise Peeling), a novel multimodal framework for molecular property prediction. Its primary objective is to explicitly disentangle the causal signal—the intrinsic molecular substructures driving a property—from spurious contextual shortcuts often learned by deep models, thereby enhancing reliability, especially under distribution shift. CLaP is built upon a batch-wise invariance principle, treating the natural feature fluctuations across mini-batches during training as contextual variation that the true causal features must ignore. The framework employs multimodal fusion, integrating three distinct molecular representations (2D topology, HELM notation, and 3D geometry) to leverage complementary structural information.

**Strengths:**

The framework effectively utilizes the principle of batch invariance, where the causal features ($Z_c$) produced by a molecule ($X$) must be able to stably predict the label ($Y$), regardless of which batch ($B$) the molecule is randomly assigned to. This constraint forces the model to shift its attention from batch-coupled spurious correlations (i.e., shortcut signals) to the sample-intrinsic, true causal structure.

**Weaknesses:**

The central weakness lies in the potential for confounding variables in the performance comparison: the reported performance gains might be primarily attributable to the richness of the input data (using three modalities: 2D, HELM, and 3D) rather than the effectiveness of the proposed Causal Layerwise Peeling (CLaP) mechanism itself. I think the authors must perform a detailed ablation study on the input modalities to cleanly isolate the contribution of the CLaP architecture.

**Questions:**

The comparison models largely use only 2D information, while CLaP uses 2D, HELM, and 3D modalities. Have the authors performed a direct comparison between CLaP and the strongest baseline, where both models use the exact same set of input modalities (e.g., 2D + 3D)?

The capacity of the trivial branch is crucial for soaking up context without contaminating the causal path. Could the authors have performed a sensitivity analysis showing how performance changes when the trivial branch capacity (e.g., number of layers or width) is varied?

---

> ### Author Response · Authors · 2025-11-21
>
> Thank you for taking the time to review our work and for the thoughtful, constructive feedback. We address each of your comments point-by-point below.(Please note that we have uploaded the revised PDF)
>
> **Weakness**
>
> >**The performance gains may come from using richer multimodal inputs rather than the CLaP mechanism itself, and a stronger modality ablation is needed to isolate CLaP’s true contribution.**
>
> We thank the reviewer for the insightful comment. To isolate the contribution of CLaP, we conducted a causal ablation study (Table 1) comparing models with and without CLaP under identical multimodal settings. The results show consistent performance gains across all nine datasets, confirming the effectiveness of the causal–trivial split itself.
>
> **Comparison: w/o causal split vs. CLaP**(Best per-dataset MAE in bold)
>
> | Method | ESOL | FreeSolv | Lipo | CycPept | HalfLife | Hepatocyte | Microsome | Solubility | Toxicity |
> |--------|-------|-----------|--------|-----------|-------------|-------------|-------------|-------------|-----------|
> | w/o causal split | 0.5094 | 1.2858 | 0.5145 | 0.3120 | 0.8301 | 1.0571 | 0.7908 | 0.7850 | 0.5184 |
> | **CLaP** | **0.4456** | **0.7020** | **0.4672** | **0.3056** | **0.7832** | **0.9520** | **0.7221** | **0.7725** | **0.5064** |
>
>
>
>
> Also, as shown in Appendix B.2, applying CLaP to a single modality yields limited improvement, which is expected — causal peeling benefits most from complementary causal signals across modalities. We hope this addresses your main concern.
>
> **Effect of causal peeling under *unimodal* vs. *multimodal* settings**
> (Best per-dataset/metric in bold)
>
> | Setting                  | ESOL MAE | ESOL MSE | FreeSolv MAE | FreeSolv MSE | Lipo MAE | Lipo MSE | CycPeptMPDB MAE | CycPeptMPDB MSE |
> |--------------------------|---------:|---------:|-------------:|-------------:|---------:|---------:|----------------:|----------------:|
> | **Multimodal (w split)** | **0.4456** | **0.3583** | **0.7020** | **0.8866** | **0.4672** | **0.3645** | **0.3056** | **0.1644** |
> | Multimodal (w/o split)   | 0.5094 | 0.4939 | 1.2858 | 2.5719 | 0.5145 | 0.4302 | 0.3120 | 0.1834 |
> | 2D Graph (w split)       | 0.9453 | 1.5993 | 1.1267 | 1.9876 | 0.5171 | 0.4506 | 0.4068 | 0.2889 |
> | 2D Graph (w/o split)     | 0.9631 | 1.6931 | 1.2003 | 2.4157 | 0.5230 | 0.4504 | 0.4077 | 0.2884 |
> | 3D Geometry (w split)    | 0.5322 | 0.5235 | 1.0640 | 1.8055 | 0.6412 | 0.6519 | 0.4221 | 0.3102 |
> | 3D Geometry (w/o split)  | 0.5219 | 0.5036 | 1.0721 | 1.9262 | 0.6260 | 0.6449 | 0.4244 | 0.3071 |
> | HELM (w split)           | -- | -- | -- | -- | -- | -- | 0.3310 | 0.2023 |
> | HELM (w/o split)         | -- | -- | -- | -- | -- | -- | 0.3194 | 0.1910 |
>
> **Questions**
>
> >**Q1:Have you compared CLaP to the strongest baselines using the **same multimodal inputs** (e.g., 2D + 3D), instead of giving CLaP more input modalities?**
>
>
> We appreciate the reviewer’s concern. We clarify that our comparisons were not limited to purely 2D modality:
>
> - **FP-GNN** already integrates **2D molecular graphs + molecular fingerprints**, representing both structural and descriptor-based modalities.
> - **MAT** incorporates **atom-level features, bond connectivity, and full 3D geometric information**, making it a **2D+3D multimodal baseline**.
> - **GSL-MPP** further extends beyond 2D by combining graph structure with **fingerprint-based molecular representations**, effectively adding a second modality.
>
> CLaP’s advantage does not stem from “extra modalities,” but from its **causal–trivial decomposition** and **invariance objectives**, which improves robustness even when strong multimodal encoders are already present.
>
>
> >**Q2:Could the model’s performance be sensitive to the capacity of the trivial branch, and is there an analysis of how varying its size affects results?**
>
> Thank you for the question. In our architecture, the causal and trivial branches are paired at every layer, and the split is applied within each block. Thus, changing the peeling depth \(L\) directly adjusts the effective capacity of both branches, including the trivial one.
>
> Our sensitivity analysis on \(L\) (Table 3 and Appendix B.4) already reflects how performance varies with different trivial-branch capacities, since increasing \(L\) increases the number of layers contributing to the trivial pathway. We observe that performance improves up to a moderate \(L\) and then saturates, indicating that the trivial branch has enough capacity to absorb contextual variation without overwhelming the causal path. In contrast, an excessively large \(L\) may lead to causal leakage and degraded performance.
>
> **Peeling Depth (L) vs. Performance on CycPeptMPDB**
>
> | L | MAE | MSE |
> |---:|------:|------:|
> | 3 | 0.3265 | 0.1903 |
> | **5** | **0.3056** | **0.1644** |
> | 7 | 0.3296 | 0.2020 |
> | 9 | 0.3297 | 0.1875 |
>
>
> **Thank you again for these thoughtful comments. We hope our responses provide helpful clarification and context for assessing our submission.**

---

### Official Review · Reviewer_Kymz · 2025-11-01

**Soundness:** 2
**Presentation:** 2
**Contribution:** 3
**Rating:** 2
**Confidence:** 4

**Summary:**

The paper proposes CLaP, a framework for interpretable regression in molecular property prediction for small molecules and cyclic peptides. The core idea is to split the representation at every layer into a “causal branch” and a “trivial branch,” and to progressively peel away batch-specific contextual signal so that only signal that is stably aligned with the target property remains. Each layer produces a causal readout that is encouraged to correlate more strongly with the target in deeper layers, while the trivial branch absorbs residual variation that is considered context-dependent rather than property-driven. The model integrates multiple types of molecular information (2D connectivity, peptide-like sequence descriptions, and 3D structure) and is trained to emphasize features that remain predictive across different batch conditions. CLaP is evaluated on four benchmark regression tasks and outperforms strong baselines. In addition, CLaP produces atom- and substructure-level saliency maps indicating which parts of a molecule are considered causal drivers of the predicted property, and these highlighted regions are reported to align with established chemical intuition, suggesting possible guidance for targeted molecular modification.

**Strengths:**

1. This work proposes a progressive, layerwise separation of causal and contextual signals. Two parallel branches are trained end to end, with their alignment to the target strengthening at deeper layers. The resulting “peeling” design contrasts clearly with invariant-subgraph heuristics while remaining easy to optimize.

2. The advantages of the model framework are demonstrated through evidences on both small molecules and cyclic peptides, which indicate transferability beyond a single regime.

3. Outputs are interpretable and chemically plausible in  atom- and substructure-level saliency maps.

**Weaknesses:**

1. The notion of “causal signal” is still operational rather than experimentally causal. The method defines causal features as those that are stable across batches and aligned with the target, and treats the rest as contextual noise. This is practical, but it is not yet demonstrated through true evidence such as editing chemistry and/or observing measured changes.

2. The proposed method relies on manually choosing a layer depth threshold 𝐿 that determines where causal “peeling” is applied. This threshold may need to be re-tuned for different model architectures and datasets, suggesting that the causal effect has a strong dependence on heuristic, experience-driven selection. In addition, training epoch progression appears to affect when the “peeled” representation emerges and stabilizes, which could further increase the sensitivity and difficulty of selecting 𝐿.

3. The interpretability results are mostly qualitative. The paper shows atom- and group-level saliency maps that align with chemical intuition, but does not yet provide a quantitative analysis where suggested edits lead to predicted directional changes in the property.

4. The paper states that the framework can be applied at every layer of the model and incorporates multi-modal gating. However, inserting this additional causal/trivial decomposition and gating at all layers may introduce nontrivial computational and memory overhead. The authors do not report these overheads or compare them to a baseline, so it is difficult to assess the scalability of the approach in larger models or fully multi-modal settings.

5. Although there is no citation for the data set, it looks the small molecular sets are from MoleculeNet, which has a lots of bias discussed in the literature.

6. The paper lack enough related work citations. the citations are also out of date, most of which are from the published work before 2022.

**Questions:**

1. How are the 3D structures obtained or selected, especially for flexible systems? If multiple conformations are possible, does the method fix one, or average over them? How sensitive is the model to noisy or low-quality 3D input, and have you compared single-modality vs multi-modality specifically on the harder peptide dataset?

2. On batch-wise invariance assumptions: The method relies on the idea that stable, property-driven signal should persist across different batch compositions, while batch-specific co-occurrences are filtered out. In practice, batching can itself introduce structure (for example, similar scaffolds ending up together). Have you examined whether the “trivial” branch is in fact capturing batch-level statistics such as scaffold frequency or label distribution, as opposed to chemically meaningful signal?

3. Corresponding to the previously mentioned weakness #2, do different model architectures or datasets require different choices of 𝐿? Is there any principled way to determine 𝐿, rather than relying on researcher intuition? In addition, the paper suggests that the “peeled” representation emerges and stabilizes over training epochs. Do you observe that the effective peeling depth drifts over training, i.e. that the same nominal 𝐿 corresponds to different behavior at different epochs? If so, does that imply that choosing 𝐿 depends not only on architecture/data but also on training stage?

---

> ### Author Response · Authors · 2025-11-21
>
> Thank you so much for your time and insights, as well as the helpful and constructive feedback! Here, we give point-by-point responses to your comments. (Please note that we have uploaded the revised PDF)
>
>
> **Weakness**
>
> >**W1: The paper’s notion of “causal signal” is not truly causal in the experimental sense.**
>
> Thank you for the insightful comment. In this work, our primary focus is on introducing the framework, establishing its consistency, and demonstrating improved performance on molecular property prediction. The consistent performance gains over strong baselines offer indirect evidence that the model is effectively separating contextual factors and preserving causal signals. We also conducted extensive case studies to validate that these signals are reasonable and indeed capture causality. While not yet fully comprehensive, we believe they already provide some meaningful causal insights.
>
> As supplementary analysis, Section 3.5 in revised version presents a counterfactual study on solubility. By replacing Cl with H and OH at high-causal-weight sites, we observe that these positions remain causally salient while the predicted solubility shifts substantially, which we believe provides evidence for meaningful molecular editing intervention. We hope these address your concern regarding the causal signal component in our framework.
>
>
> >**W2: The method requires manually tuning the peeling depth L, making it sensitive to heuristics and dataset/model choices.**
>
> We appreciate the reviewer’s concern. In our framework, the peeling depth **L** functions as a structural hyperparameter, similar to the number of layers in transformers or GNNs, and is selected empirically per dataset.
>
> Importantly, training dynamics are not left uncontrolled: besides correlation schedule loss for the causal branch, we also introduce a **monotonicity constraint** (Eq. 9) that enforces layer-wise correlation improvement. This encourages stable refinement of causal representations across depths, reducing the sensitivity to the specific choice of **L**.
>
>
>
>
> >**W3: Interpretability is demonstrated only qualitatively.**
>
> Thank you for pointing this out. We agree that additional quantitative validation would further strengthen our interpretability claims. However, the primary contribution of this work is the introduction of the causal–trivial peeling framework itself. To provide supporting evidence of generated causal weight, in a counterfactual study, where we replace the original high–causal-weight low-solubility atom with alternative high-solubility atoms. The results show that the causal weights shift directionally in a manner consistent with solubility changes, demonstrating that the model responds appropriately to counterfactual interventions rather than merely relying on surface correlations. We hope this offers a more realistic demonstration of how the causal signals can be applied in practice.
>
> >**W4: The method may add nontrivial compute and memory overhead, but no efficiency or scalability analysis is provided.**
>
> We appreciate the reviewer’s concern. Each causal block adds only lightweight linear projections and elementwise gating, with cost scaling as **O(N·D)** per modality — comparable to a single GNN layer update. The fusion gate introduces a 3-way softmax over modality-specific embeddings, adding **O(M·D)** compute per layer, which is negligible relative to backbone message-passing operations. The trivial branch reuses the same dimensionality and consists of a single projection head, adding **O(D)** parameters and no additional graph operations.
> N = number of atoms, M = number of modality, D = embedding dimension,
>
> Memory usage increases linearly with the number of layers due to storing per-layer scalar readouts, but this overhead is minimal compared to node embeddings and attention states, and does not meaningfully affect scalability.

---

> > ### Author Response · Authors · 2025-11-21
> >
> > >**W5 and W6: The dataset appears to come from MoleculeNet, which is known to contain biases and lacks proper citation.**
> >
> > Thank you for pointing this out. We have added the appropriate citations for all datasets used.
> >
> > We are also aware that you may be referring to [1], which points out that small-molecule benchmarks may not guarantee true generalization due to limited chemical space coverage. Nonetheless, these datasets are widely used in molecular property prediction research and still remain standard evaluation benchmarks.
> >
> > As a supplement,, in our main results including additional 5 OOD datasets and three most recent baseline models \[2, 3, 4\]  in the revised version, our method consistently outperforms others under identical training conditions even in OOD setting, which further demonstrates the generalizability of our approach. We hope this addresses your concern about the dataset and baselines we included.
> >
> > **Comparison across nine benchmarks**
> > (Best per-dataset MAE in bold)
> >
> > | Method | ESOL | FreeSolv | Lipo | CycPept | HalfLife | Hepatocyte | Microsome | Solubility | Toxicity |
> > |--------|-------|-----------|--------|-----------|-------------|-------------|-------------|-------------|-----------|
> > | CAL | 0.4923 | 1.1273 | 0.6217 | 0.3682 | 1.0189 | 0.9839 | 0.8335 | 1.0572 | 0.6636 |
> > | FP-GNN | 0.5039 | 1.1561 | 0.4809 | 0.3487 | 0.8857 | 1.0836 | 0.8083 | 0.8383 | 0.5242 |
> > | CGR | 1.5853 | 2.7268 | 0.9670 | 0.5792 | 1.0360 | 1.2371 | 1.2544 | 1.6741 | 0.7794 |
> > | GSL | 1.0032 | 2.3476 | 0.9007 | 0.6590 | 0.9365 | 1.1045 | 1.0723 | 1.5883 | 0.7414 |
> > | DIR | 0.6267 | 2.2420 | 0.6328 | 0.4243 | 0.9600 | 1.0780 | 1.1580 | 1.1890 | 0.6810 |
> > | MAT | 0.5394 | 0.7680 | 0.5284 | 0.4129 | 0.9915 | 1.5660 | 1.1350 | 0.9250 | 0.8622 |
> > | MolFCL | 0.4660 | 1.0300 | **0.4330** | 0.3690 | 0.9210 | 1.1160 | 0.9430 | **0.6990** | 0.7530 |
> > | **CLaP** | **0.4456** | **0.7020** | 0.4672 | **0.3056** | **0.7832** | **0.9520** | **0.7221** | 0.7725 | **0.5064** |
> >
> >
> > \[1\] Kretschmer, F., Seipp, J., Ludwig, M. et al.
> > Coverage bias in small molecule machine learning.
> > *Nature Communications*, 16, 554 (2025).
> >
> > \[2\] Xiang Tang, Qichang Zhao, Jianxin Wang, Guihua Duan.
> > MolFCL: predicting molecular properties through chemistry-guided contrastive and prompt learning.
> > Bioinformatics, 41(2), 2025.
> >
> > \[3\] Bangyi Zhao, Weixia Xu, Jihong Guan, Shuigeng Zhou.
> > *Molecular property prediction based on graph structure learning.*
> > Bioinformatics, 40(5), 2024.
> >
> > \[4\] Yujia Yin, Tianyi Qu, Zihao Wang, Yifan Chen.
> > *A Recipe for Causal Graph Regression: Confounding Effects Revisited.*
> > ICML 2025.
> >
> >
> >
> >
> > **Question**
> >
> > >**Q1: It is unclear how 3D conformations are chosen or handled for flexible molecules, how robust the model is to noisy 3D input, and whether single vs. multimodal performance is compared on the harder peptide dataset.**
> >
> > Thank you for raising these points. In our setting, 3D structures apply ETKDG (up to 5 initial conformers), and energy-minimize the first embedded conformer using UFF. We acknowledge that flexible systems (e.g., cyclic peptides) may adopt multiple conformations. In this work, we fix one minimized conformer instead of averaging over multiple. However, CLaP remains robust because the 3D representation is only one modality, and our layerwise gating selectively allocates lower weights to modalities that contribute less to the predictive signal.
> >
> > As shown in our unimodal vs. multimodal ablation on CycPeptMPDB (Table 5), the 3D-only model performs worse than SM-only or PE-only models. However, combining modalities yields substantially better performance. This is expected, as causal splitting leverages complementary causal information across modalities, allowing the model to fully exploit causal information that may not be present in any single view.
> >
> > **Effect of causal peeling under different modalities in CycPeptMPDB:**
> >
> > | Setting                  | MAE     | MSE     |
> > |--------------------------|--------:|--------:|
> > | **Multimodal (w split)** | **0.3056** | **0.1644** |
> > | Multimodal (w/o split)   | 0.3120 | 0.1834 |
> > | 2D Graph (w split)       | 0.4068 | 0.2889 |
> > | 2D Graph (w/o split)     | 0.4077 | 0.2884 |
> > | 3D Geometry (w split)    | 0.4221 | 0.3102 |
> > | 3D Geometry (w/o split)  | 0.4244 | 0.3071 |
> > | HELM (w split)           | 0.3310 | 0.2023 |
> > | HELM (w/o split)         | 0.3194 | 0.1910 |

---

> > > ### Author Response · Authors · 2025-11-21
> > >
> > > >**Q2: The method assumes batch-wise invariance, but batching itself may introduce structure. It's unclear whether the trivial branch is capturing true non-causal noise or merely batch-level scaffold and label biases.**
> > >
> > > Thank you for the question. For our framework design’s trivial part, we provide **empirical evidence** that the model separates trivial factors successfully:
> > > 1. **Removing the trivial branch (Table 1)**
> > > Performance drops consistently also in OOD scaffold-split datasets, showing the trivial branch is necessary for absorbing batch-dependent variation rather than causal signals.
> > >
> > > **Comparison: w/o context vs. CLaP**
> > > (Best per-dataset MAE in bold)
> > > | Method | ESOL | FreeSolv | Lipo | CycPept | HalfLife | Hepatocyte | Microsome | Solubility | Toxicity |
> > > |--------|-------|-----------|--------|-----------|-------------|-------------|-------------|-------------|-----------|
> > > | w/o context | 0.4732 | 1.0753 | 0.5824 | 0.3214 | 0.8123 | 1.0330 | 0.7805 | 0.7801 | 0.5189 |
> > > | **CLaP** | **0.4456** | **0.7020** | **0.4672** | **0.3056** | **0.7832** | **0.9520** | **0.7221** | **0.7725** | **0.5064** |
> > >
> > > 2. **Re-batching intervention (Table 3)**
> > >    Causal representations remain stable under changed batch composition, while trivial factors shift—demonstrating intended invariance.
> > >
> > > 3. **Interpretability case study (Fig. 4)**
> > > Causal saliency highlights chemically meaningful atoms (N, O), while scaffolds receive low and equal attribution—consistent with trivial information. Although we did not conduct a full quantitative human evaluation, multiple analyzed cases align well with established chemical intuition.
> > >
> > >
> > >
> > > >**Q3: Does the optimal peeling depth L depend on architecture, dataset, or training stage? And is there a principled way to choose L beyond intuition?**
> > >
> > >
> > > Thank you for the question. As noted earlier, we treat the peeling depth **L** as a standard architectural hyperparameter. Different architectures or datasets may prefer different values, and we select **L** based on validation performance rather than intuition. And we do not observe drift in the effective peeling depth during training. Once **L** is chosen, the model remains stable, and no dynamic adjustment is needed. In practice, we simply choose the **L** that best captures the causal signal on the validation set, making the framework robust without depending on training stage behavior.
> > >
> > > **We appreciate the reviewers’ time and constructive feedback! We hope our clarifications address your concerns and help in evaluating our work.**

---

### Meta-Review · Area_Chair_xXpy · 2026-01-05

**Summary:**

This paper proposes CLaP, a layerwise framework for multimodal molecular property prediction that aims to separate “causal” signals from contextual or spurious correlations using batch-wise invariance and a causal–trivial split at each layer. Reviewers generally agreed that the paper is technically sound, clearly structured, and supported by extensive experiments, including ablations and additional benchmarks added during rebuttal. The empirical results consistently show improvements over selected baselines, and the interpretability outputs are plausible and well presented.

At the same time, reviewers raised substantive concerns about whether the paper’s causal framing and overall contribution meet the bar for ICLR. In particular, reviewers questioned the definition of causality used in the paper, the reliance on heuristic design choices, and the difficulty of disentangling the proposed causal peeling mechanism from the effects of multimodal inputs and regularization. Although the authors provided detailed rebuttals and additional experiments, no reviewer replied after the rebuttal to indicate that their concerns had been resolved or that they intended to change their score. Taking into account the remaining concerns and the overall review signal, the paper is recommended for rejection.

**Reviewer Concerns:**

Across reviews, a recurring concern is that the notion of “causal signal” in this work is defined operationally through model behavior, rather than grounded in experimental or interventional causality. The method defines causal features as those that are stable across batches and aligned with the target, which is a reasonable invariance-based modeling assumption, but reviewers were not fully convinced that this corresponds to true causal structure rather than stable correlations induced by dataset composition, batching effects, or benchmark bias.

Reviewers also expressed concerns about the reliance on heuristic choices, such as the manually selected peeling depth, and the sensitivity of the method to architectural and training decisions. While the authors argue that this is similar to choosing other architectural hyperparameters, reviewers felt that the causal interpretation depends strongly on these choices and lacks a principled or data-driven justification. In addition, several reviewers noted that it remains difficult to isolate the contribution of the CLaP mechanism itself from the benefits of using richer multimodal inputs and additional losses, despite the ablations added during rebuttal.

Further concerns include limited discussion of computational overhead and scalability in fully multimodal or larger-scale settings, as well as the generality of the approach beyond the evaluated regression benchmarks. Although the authors responded carefully to these points and expanded the experimental section, these responses did not fully address the underlying conceptual concerns. No reviewer followed up after the rebuttal to indicate that their concerns had been alleviated or that they would revise their score. After carefully reading the paper and the author's rebuttal, I believe the aforementioned concerns remain.

**Reviewer Scores:**

The reviewer scores were 2, 4, 4, and 4, reflecting a generally below-threshold evaluation. Reviewers acknowledged the technical quality of the work and the consistency of the empirical improvements, but expressed reservations about the causal framing, experimental scope, and overall contribution level. As no reviewer followed up, no reviewer explicitly indicated an intention to change their score following the rebuttal or discussion.

---

### Decision · Program_Chairs · 2026-01-26

Reject